# Superflow decay in a toroidal Bose gas: The effect of quantum and thermal fluctuations

Z. Mehdi[1*], A. S. Bradley[2], J. J. Hope[1], S. S. Szigeti[1]

**1** Department of Quantum Science, Research School of Physics, The Australian National University, Canberra 2601, Australia
**2** The Dodd-Walls Centre for Photonic and Quantum Technologies, Department of Physics, University of Otago, Dunedin, New Zealand
* zain.mehdi@anu.edu.au

May 7, 2021

## Abstract

We theoretically investigate the stochastic decay of persistent currents in a toroidal ultra-cold atomic superfluid caused by a perturbing barrier. Specifically, we perform detailed three-dimensional simulations to model the experiment of Kumar *et al.* in [Phys. Rev. A **95** 021602 (2017)], which observed a strong temperature dependence in the timescale of superflow decay in an ultracold Bose gas. Our *ab initio* numerical approach exploits a classical-field framework that includes thermal fluctuations due to interactions between the superfluid and a thermal cloud, as well as the intrinsic quantum fluctuations of the Bose gas. In the low-temperature regime our simulations provide a quantitative description of the experimental decay timescales. At higher temperatures, our simulations give decay timescales that range over the same orders of magnitude observed in the experiment, however, there are some quantitative discrepancies. In particular, we find a much larger perturbing barrier strength is required to simulate a particular decay timescale (between $\sim 0.15\mu$ and $\sim 0.5\mu$), as compared to the experiment. We rule out imprecise estimation of simulation parameters, systematic errors in experimental barrier calibration, and shot-to-shot atom number fluctuations as causes of the discrepancy. However, our model does not account for technical noise on the trapping lasers, which may have enhanced the superflow decay in the experiment. For the intermediate temperatures studied in the experiment, we also observe some discrepancy in the sensitivity of the decay timescale to small changes in the barrier height, which may be due to the breakdown of our model's validity in this regime.

# 1   Introduction

Ultracold atomic Bose gases are versatile, highly configurable systems, in part due to their isolation from environmental effects, precise controllability with magnetic, optical, and rf fields, and the accessible imaging of many atomic observables [1, 2]. This makes these systems ideal platforms for experimentally investigating superfluidity and many-body quantum phenomena [3–6]. In particular, atomic Bose-Einstein condensates (BECs) confined to multiply-connected geometries such as a toroid are exceptionally well-suited for investigating persistent currents of superfluid flow [2, 7, 8], which may provide insights into the nature of supercurrents in superconducting materials. The first experimental demonstrations of persistent flow in a toroidal BEC were performed more than a decade ago [9, 10]. Since then, experiments with superfluid toroidal BECs have investigated the creation and stability of persistent currents [11–15], atomic-gas analogs of quantum phenomena in electronic devices [16, 17], quantum field dynamics in cosmic inflation [18], and compact atom interferometry in optical waveguides [19]. There have also been many recent theoretical works that have investigated superfluidity and persistent currents in one-dimensional systems [20–25], protocols for atomic-gas superfluid circuits [26, 27], superflow

in dipolar supersolids [28], and mechanisms for superflow decay, both within mean-field theory [29–31] and beyond [32, 33].

Superfluid studies in toroidal atomic gases are particularly relevant to the emerging field of atomtronics, which broadly aims to develop circuit-based atomic gas devices [34, 35]. Toroidal superfluids could be used realize the matter-wave equivalent of a superconducting quantum-interference device (SQUID) [36]. However, a robust, well-functioning atomtronic SQUID requires precision control over the superflow current at the single-quantum level - as indeed does almost any atomtronic device based on superfluidity. Constructing theoretical models capable of quantitatively describing superfluid experiments is therefore essential for the future development of increasingly sophisticated atomtronic devices [37].

Here, we focus on one critical aspect of superflow: the lifetime of persistent current states in the presence of a repulsive barrier. There have been several experiments where a weak-link perturbing barrier was used in toroidal BECs to create persistent current states and alter their stability [10–13, 38, 39]. However, the experimentally-measured critical velocity of superflow significantly differs to the predictions of mean-field theory [10, 39]. Indeed, despite numerous theoretical investigations into the underlying mechanisms of superflow decay in the presence of a perturbing barrier [26, 29–33, 40], a detailed understanding of the nature and origin of superflow instability remains lacking.

Recently, an experiment by Kumar *et al.* studied the temperature dependence of superflow decay in a toroidal atomic superfluid [15]. Their experiments found that the rate of superflow decay was strongly dependent on temperature and quantitatively disagreed with both quantum tunnelling through an energy barrier and thermal activation over an energy barrier[1]. A subsequent theoretical analysis found that the experimental results could also not be reproduced with a model of thermally-activated phase slips within mean-field theory [31]. These findings, combined with the theoretical analysis [32] of a related experiment by Ramanathan *et al.* [10], suggests that a more sophisticated theoretical framework is needed to quantitatively model superflow decay in a toroidal BEC.

In this work, we perform detailed three-dimensional simulations to model the experiment of Kumar *et al.* [15]. Our model is constructed within classical field (c-field) methodology, a well-developed framework for quantitatively describing the non-equilibrium dynamics of dilute Bose gases in the quantum degenerate regime [41]. In particular, our model goes beyond mean-field theory and includes both the inherent fluctuations of the quantum state and finite-temperature interactions with an incoherent thermal reservoir. The latter is essential in order to describe the strong dependency of superflow decay on temperature observed in the experiment. Notably our model does not contain any fitted parameters, with all simulation parameters determined *ab initio* from the experimental atom numbers and temperatures.

Our simulations are able to capture both qualitative and quantitative features of the experiment, with quantum and thermal fluctuations leading to a stochastic decay of the superflow. We calculate the timescale of the decay and compare this to experimental values. The computed decay timescales range over the same orders of magnitude as the experiment and we see quantitative agreement for the lowest temperature studied. However, at higher temperatures the simulations require a larger perturbing barrier height than the experiment to achieve the experimentally-observed decay timescales. This discrepancy is largest for the highest temperature studied in the experiment, where the validity of our model is well established. This suggests that there is some aspect of the experiment that is not captured in our model. Although we have explored some possibilities for the noted

---

[1]The energy barrier used in these models assumed that a solitonic vortex was the lowest-energy excitation capable of coupling quantized circulation states.

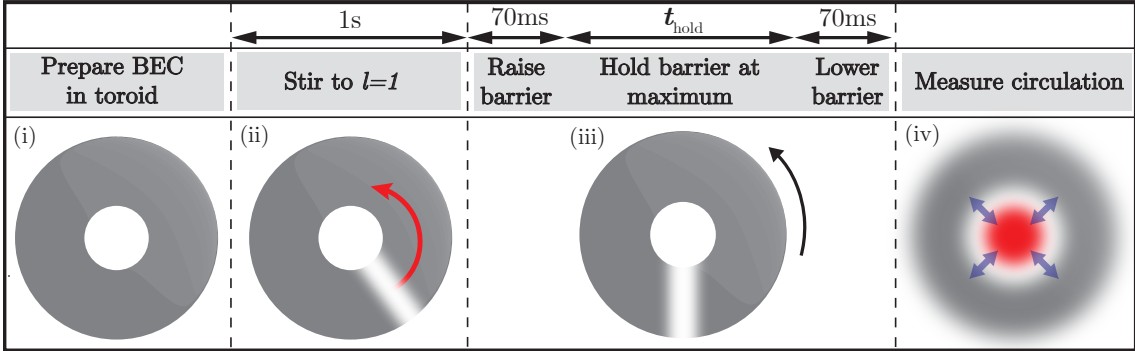

Figure 1: Schematic of the key steps in the experimental procedure of Ref. [15]. (i) The atomic cloud is first prepared in a toroidal trap at temperature $T$. (ii) The condensate is then prepared in the $l = 1$ circulation state by stirring a barrier around the condensate. (iii) To induce decay of the superflow, a barrier with strength weaker than the chemical potential is raised over a period of 70ms, held constant for time $t_{\text{hold}}$, and then lowered over 70ms. (iv) A measurement of the circulation is then made by releasing the atoms from the toroidal trap and subsequently observing their interference with an auxiliary disk of atoms in the center (pictured in red).

discrepancies in this work, the precise origin of this effect remains unclear.

## 2    Details of the experiment

Here we briefly describe the experimental procedure of Ref. [15]. The key steps are summarised schematically in Fig. 1. In the experiment, ultracold $^{23}$Na atoms were confined in a toroidal optical trapping potential with a mean radius of $r_0 = 22.4\mu$m. Their experiment considered Bose gases prepared at four different temperatures: $T = 30, 40, 85$ and 195 nK. For the lowest two temperatures, vertical trapping was provided by a blue-detuned beam. For the higher temperatures, the vertical confinement was provided by a red-detuned beam, with atoms residing in the region of greatest light intensity.

To prepare the atomic superfluid in the first quantized circulation state ($l = 1$), a weak-link barrier slightly stronger than the chemical potential was raised adiabatically, stirred around the condensate as depicted in Fig. 1(ii), and then adiabatically lowered. In total the stirring procedure took $\sim$1s and prepared the desired circulation state with a fidelity of roughly 96%. The barrier itself was generated by rapidly scanning a Gaussian beam across the radial extent of the condensate, the time-average of which is approximately constant over the condensate density.

To induce decay of the superflow from the $l = 1$ circulation state to a non-circulating state ($l = 0$), the experiment introduced a stationary perturbing barrier with peak height weaker than the BEC's chemical potential. This barrier height was raised linearly over a period of 70ms, held at its maximum height $V_b$ for some variable time $t_{\text{hold}}$ between 0.2 s and 4.6 s, and then lowered linearly over another 70ms period. In order to keep the total time of the experiment constant at $\sim$7s, there was a variable time delay between the stirring stage and raising the barrier.

To measure the circulation state, the BEC was released from the toroidal trap and allowed to interact with a reference disk of atoms ($l = 0$) held at the center of the trap. The winding number $l$ was then extracted by observing the interference between the toroidal BEC and disk atoms. The measurement works as follows: if the BEC is in an

$l = 1$ persistent current state when it is released from the trap, then the 'hole' in the atomic density at the center of the cloud remains even as the cloud falls and expands. Consequently, the atomic cloud does not significantly overlap with the reference disk of atoms, yielding no measureable interference. In contrast, if the superflow had decayed to the $l = 0$ state, the hole in the atomic cloud fills during free expansion, resulting in an interaction between the released atoms and the reference disk, and therefore a measureable interference pattern. For a given set of parameters $\{T, V_b, t_{\text{hold}}\}$, the measurement was repeated $16-18$ times to calculate the mean winding number $\langle l \rangle$.

The decay timescale $\tau$ was computed by varying $t_{\text{hold}}$ while keeping $T$ and $V_b$ fixed, and then fitting $\langle l \rangle$ to an exponential model. Given the finite number of measurements and the range of $t_{\text{hold}}$ values considered in the experiment, there was only a finite range of $\tau$ that could be distinguished from infinitely fast decay ($\tau = 0$) or no decay ($\tau = \infty$). The largest value of $\tau$ distinguishable from $\tau = \infty$ corresponds to the case where, for the largest value of $t_{\text{hold}}$, only one of the 18 measurements registers a decay event. Similarly, the smallest value of $\tau$ distinguishable from $\tau = 0$ corresponds to the case where, for the smallest value of $t_{\text{hold}}$, all but one of the 18 measurements registers a decay event. This sets limits on the possible values of $\langle l \rangle$, giving the range of $\tau$ values measurable by the experiment as $70 \text{ ms} \lesssim \tau \lesssim 80 \text{ s}$.

## 3 Theoretical model

Decay of superflow due to the presence of a perturbing barrier is an inherently out-of-equilibrium scenario, for which there are limited theoretical tools capable of capturing both quantum and thermal effects. Our model of the experiment reported in Ref. [15] is formulated within the c-field theoretic framework, which is inherently non-perturbative and therefore well suited to studying a range of out-of-equilibrium phenomena, at both zero and finite temperature [41]. In this section we describe both our c-field model and our numerical simulation procedure.

### 3.1 Classical field methodology

The essential idea of c-field methods is that the macroscopically occupied modes of a degenerate Bose gas can be well described by an equation of motion for a classically-valued field $\psi$. Formally, c-field theories are constructed by dividing the full quantum field theory into a low-energy band $\mathbf{C}$, which contains all modes of high occupation, and a high-energy band $\mathbf{I}$ which contains the remaining sparsely-occupied modes. This leads to a decomposition of the field operator as:

$$\hat{\psi} = \hat{\psi}_{\mathbf{C}} + \hat{\psi}_{\mathbf{I}} \,, \tag{1}$$

where $\hat{\psi}_{\mathbf{C}}$ and $\hat{\psi}_{\mathbf{I}}$ are field operators for the $\mathbf{C}$ and $\mathbf{I}$ regions, respectively. An energy cutoff $\epsilon_{\text{cut}}$ defines the division of the field theory into $\mathbf{C}$ and $\mathbf{I}$ regions and a projector $\mathcal{P}$ ensures the two regions remain separated dynamically, i.e. $\mathcal{P}\{\hat{\psi}\} = \hat{\psi}_{\mathbf{C}}$. Classical field theories treat $\hat{\psi}_{\mathbf{C}}$ as a classical field $\psi$, and thus neglect the discrete nature of the atoms within the $\mathbf{C}$ region. In contrast, the $\mathbf{I}$ region is treated as a static thermal reservoir. The dynamics of the c-field $\psi$ can be determined via a phase-space correspondence that maps the equations of motion for $\hat{\psi}_{\mathbf{C}}$ to equations of motion for $\psi$ [41]. Formally, within the phase-space framework, $\psi$ is a stochastic sample of the $\mathbf{C}$ region's approximate phase-space distribution, with expectations of physical quantities given by ensemble averages of moments of $\psi$. However, an individual sample of $\psi$ can often be loosely interpreted as the

outcome of a single experimental run where, for example, the density of the Bose gas is $|\psi|^2$ [42]. For further details regarding c-field methodology, and examples of applications to non-equilibrium phenomena in Bose gases, see Ref. [41] and references therein.

Classical field methods have successfully modelled ultracold Bose gases at both zero and finite temperature [41, 43]. For systems near zero temperature, the occupation of the **I** region is negligible and thus the dominant beyond-mean-field effect is often inherent quantum fluctuations of the Bose gas. Within this regime, *zero-temperature truncated Wigner (TW)* [44–46] is the dominant c-field approach. For the closed-system dynamics typical of many BEC experiments, $\psi$ is governed by a Gross-Pitaevskii equation (GPE) with initial conditions sampled from the Wigner distribution of the initial state. For many $T = 0$ non-equilibrium phenomena it is sufficient to treat this initial condition as a multimode coherent state that is sampled by seeding the initial mean-field condensate wavefunction with on average half an atom of vacuum noise per mode [47]. Zero temperature TW with this initial condition has successfully modelled BEC dynamics in regimes where nonclassical particle correlations become important [48–57]. For finite-temperature studies, the relevant c-field theory is the *stochastic projected Gross-Pitaevskii equation* (SPGPE), which describes interactions between degenerate modes of the quantum field with a static thermal reservoir [58]. The SPGPE and its sub-theories have been used extensively to study Bose gases both in and out of equilibrium, such as in Refs. [59–61] and Refs. [41, 62–77], respectfully. Notably, the SPGPE has been able to quantitatively, describe experimental results, such as in Refs. [59, 78–81]. The SPGPE is typically applied to systems with a large thermal fraction, usually at temperatures ranging from $T{\sim}T_c/2$ (where $T_c$ is the critical temperature of condensation) to just over $T{\gtrsim}T_c$.

The experiment of Kumar *et al.* that we model in this work investigated superflow decay in the presence of a relatively small thermal cloud; specifically, the experiment studied atomic superfluids at temperatures between $T = 30\text{nK}$ ($\sim0.05T_c$) and $T = 195\text{nK}$ ($\sim0.4T_c$) [15]. Unfortunately this is outside the typical regime of validity for both zero-temperature TW, which assumes a negligible thermal cloud, and the SPGPE, which assumes a larger proportion of thermal atoms. This is a low-temperature regime where neither quantum or thermal fluctuations truly dominate over one another. To address this regime, our model combines zero-temperature TW and SPGPE theory to include both quantum and thermal fluctuations, following the approach outlined in the Supplemental Materials of Ref. [57]. In our model, initial states are first sampled from the grand canonical ensemble of a Bose gas at thermal equilibrium using the SPGPE. Quantum fluctuations are then included by adding half a quantum of vacuum noise to each mode of the sample, as is done when sampling a coherent state in TW [47]. These initial states are then evolved using the SPGPE, with parameters describing reservoir interactions estimated from the atom number and temperature reported in the experiment. This model reduces to zero-temperature TW at very low temperatures, and to SPGPE at higher temperatures, both of which are expected to be quantitative models in their regimes of validity.

Additionally, a quantitative description of Ref. [15] requires the three-dimensional form of the SPGPE. This is because the vertical confinement in the experiment, $\omega_z$, is not sufficiently large that all excitations in the $z$ dimension are suppressed. Specifically, the trapping parameters of the experiment do not satisfy the condition $\hbar\omega_z \gg \mu$, which is required for an effective two-dimensional model to be a quantitatively correct description of the Bose gas.

## 3.2 SPGPE Theory

The simple-growth stochastic projected Gross-Pitaevskii equation can be written as:

$$i\hbar d\psi = \mathcal{P}\big\{(1 - i\gamma)(\mathcal{L} - \mu)\psi dt + i\hbar d\xi_\gamma(\mathbf{r}, t)\big\},\tag{2}$$

where $\mu$ is the chemical potential of the reservoir, $d\xi_\gamma(\mathbf{x}, t)$ is a complex Gaussian noise of mean zero and correlation

$$\mathbb{E}[d\xi_\gamma^*(\mathbf{r})d\xi_\gamma(\mathbf{r}')] = 2\gamma\frac{k_{\mathrm{B}}T}{\hbar}\delta^{(3)}(\mathbf{r} - \mathbf{r}')dt,\tag{3}$$

and

$$\mathcal{L} = -\frac{\hbar^2}{2m}\nabla^2 + U_t(r, z) + U_b(r, \theta, z) + g|\psi(r, \theta, z)|^2\tag{4}$$

is the Gross-Pitaevskii mean-field operator in cylindrical coordinates. Here $g = 4\pi\hbar^2 a_s/m$ is the atom-atom interaction strength where $a_s \approx 52a_0$ for $^{23}$Na, which was the atomic species used in the experiment. The term $U_t(r, z)$ corresponds to the toroidal trapping potential, which is approximately harmonic:

$$U_t(r, \theta, z) = \frac{1}{2}m\left(\omega_r^2(r - r_0)^2 + \omega_z^2 z^2\right),\tag{5}$$

where $r_0 = 22.46\mu$m is the mean radius of the ring in the experiment of Kumar *et al.* The form of the SPGPE we use in this work, Eq. (2), neglects number-conserving scattering interactions between the **C** and **I** regions, which are often referred to as the 'scattering' or 'energy-damping' terms [82]. These terms are commonly neglected in studies with SPGPE under the assumption that they do not significantly contribute to system dynamics near equilibrium [41]. We have confirmed that these terms have little quantitative effect on the results of this work (see Appendix D).

The projector $\mathcal{P}$ restricts the c-field $\psi$ to the low-energy subspace **C** defined by $\epsilon_{\mathrm{cut}}$. This energy cutoff is typically chosen such that the highest-energy single-particle modes contained within the **C** region have an occupation of roughly $\overline{n}_{\mathrm{cut}}\sim 1-10$. We choose the energy cutoff by inverting the Bose-Einstein distribution for a fixed $\overline{n}_{\mathrm{cut}}$ [83]:

$$\epsilon_{\mathrm{cut}} = \ln\left(1 + \frac{1}{\overline{n}_{\mathrm{cut}}}\right)k_B T + \mu.\tag{6}$$

In all our calculations, we fix $\overline{n}_{\mathrm{cut}} = 1$, which gives an average occupation of $n \approx 1$ for single-particle modes near the cutoff (see Fig. 11 in Appendix B).

The dimensionless damping strength $\gamma$ in Eq. (2) gives the rate of reservoir interactions, and characterises the speed at which the c-field approaches thermal equilibrium with the thermal reservoir. It can be *a priori* determined from the chemical potential of the reservoir $\mu$, the reservoir temperature $T$, and the energy cutoff $\epsilon_{\mathrm{cut}}$ [62]:

$$\gamma = \frac{8a_s^2}{\lambda_{\mathrm{dB}}^2}\sum_{j=1}^{\infty}\frac{e^{\beta\mu(j+1)}}{e^{2\beta\epsilon_{\mathrm{cut}}j}}\Phi[e^{\beta(\mu-2\epsilon_{\mathrm{cut}})}, 1, j],\tag{7}$$

where $\beta = 1/(k_{\mathrm{B}}T)$, $\lambda_{\mathrm{dB}} = \sqrt{2\pi\hbar^2/(mk_{\mathrm{B}}T)}$ is the thermal de Broglie wavelength, and $\Phi[z, x, a]$ is the Lerch transcendent. If $\gamma = 0$ and the projector is neglected, then Eq. (2) becomes the Gross-Pitaevskii equation (GPE).

We numerically implement Eq. (2) in a basis of approximate single-particle modes of the toroidal trap, extending the approach outlined for the projected GPE in Ref. [84]. A notable benefit of this approach is that the energy cutoff can be consistently defined in this

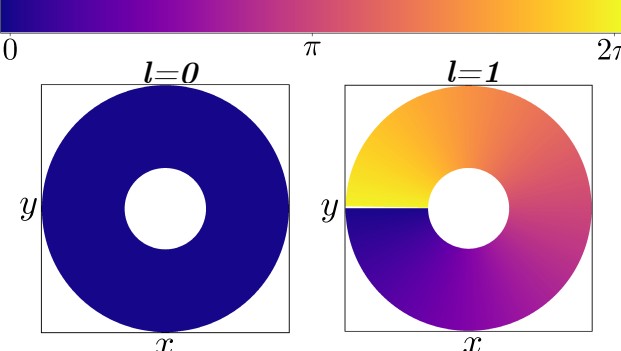

Figure 2: A visualisation of exemplary phase profiles in the $z = 0$ plane in a toroidal geometry, for quantized circulation states with winding number $l = 0$ (left) and $l = 1$ (right). For the $l = 1$ circulation state, the phase increases by $2\pi$ as it winds around the central hole. The phase is only well defined in the region where there is significant superfluid density, and thus the center of the torus is omitted in this image.

single-particle basis, which diagonalises the Hamiltonian at high energies. Furthermore, only a relatively small number of single-particle modes are required in order to represent the **C** region, compared to the much larger number of points required to represent the c-field on a three-dimensional cartesian grid. This enables the detailed finite-temperature 3D simulations performed in this work which, given the long decay timescales observed in the experiment, are impracticable using grid-based methods. Details of our simulation implementation are given in Appendix A. A description of the simulation parameters, and a justification of the parameter values chosen, is found in Appendix B. Unless otherwise stated, SPGPE simulation results reported in this work are averages over an ensemble of 96 trajectories. All reported observables are accompanied by a 95% confidence interval, estimated as twice the standard error of the ensemble averages.

### 3.3 The winding number

It is well known that the circulation $\mathcal{C}$ around some closed loop $D$ in a superfluid is quantized in integer multiples of $2\pi\hbar/m$ [85]:

$$\mathcal{C} = \oint_D \vec{v} \cdot \vec{dl} = \frac{2\pi\hbar}{m} l \,, \tag{8}$$

where $\vec{v} = (v_r, v_\theta, v_z)^{\mathsf{T}}$ is the superfluid velocity field in cylindrical coordinates and the *winding number* $l$ is an integer. The quantization of circulation is due to the relationship between velocity and the phase of the superfluid order parameter, as the phase must change by an integer multiple of $2\pi$ after 'winding' around the loop $D$.

For circulation around the annulus of a toroidal superfluid, the winding number is a *topological* quantity. This is because the torus is a multiply-connected topological space, which allows winding of the superfluid phase without the existence of vortices. In Fig. 2, we show exemplary phase profiles for the zero circulation state ($l = 0$), and the $l = 1$ circulation state with $2\pi$ phase winding.

In our numerical calculations, we compute $l$ in each stochastic trajectory of the SPGPE and take the ensemble average to get the average winding number $\langle l \rangle$. For convenience of calculation, we choose a loop around the trap minimum, which gives the following expression for the winding number (in cylindrical coordinates):

$$\langle l \rangle = \frac{m r_0}{2\pi\hbar} \mathbb{E} \left[ \int_0^{2\pi} d\theta \, v_\theta(r = r_0, \theta, z = 0) \right] \,, \tag{9}$$

where $\mathbb{E}[*]$ denotes an ensemble average over stochastic trajectories. In any given trajectory, we calculate the angular velocity field $v_\theta$ by dividing the particle current by the density:

$$v_\theta(r, \theta, z) = \frac{j_\theta(r, \theta, z)}{n(r, \theta, z)} = \frac{\hbar}{mr} \frac{\text{Im}\{\psi(r, \theta, z)^* \partial_\theta \psi(r, \theta, z)\}}{|\psi(r, \theta, z)|^2} \, . \tag{10}$$

## 3.4 Initial state generation

The initial conditions for our simulations of Eq. (2) are stochastic samples of the **C** region's quantum state at thermal equilibrium in the grand canonical ensemble. It is common for finite-temperature studies in the c-field framework to neglect quantum fluctuations in the initial state, on the assumption that they are dominated by thermal fluctuations in the high-temperature regime where SPGPE is typically applied. However, this assumption is not appropriate for the regime studied in Kumar *et al.*, where neither quantum and thermal fluctuations dominate over the other. To include both quantum and thermal effects, we add half an atom of vacuum noise per mode to initial samples of the grand canonical ensemble. This approach is similar to that used in Ref. [86] and the Supplemental Material of Ref. [57], and is akin to sampling the Wigner distribution of an incoherent mixture of coherent states that reproduce the statistics of the grand canonical ensemble. Specifically,

$$\psi(t = t_0) = \phi_\text{therm} + \frac{1}{\sqrt{2}} \mathcal{P}\{\eta\} \tag{11}$$

where $\phi_\text{therm}$ is the complex amplitude of a multimode coherent state $|\phi_\text{therm}\rangle$, sampled such that $\hat{\rho} = \mathbb{E}[|\phi_\text{therm}\rangle \langle\phi_\text{therm}|]$ is the density matrix for a thermal equilibrium state in grand canonical ensemble[2], and $\eta$ is a complex Gaussian noise satisfying the correlation

$$\mathbb{E}[\eta(\mathbf{r})^* \eta(\mathbf{r}')] = \delta^{(3)}(\mathbf{r} - \mathbf{r}') \, . \tag{12}$$

Each sample $\phi_\text{therm}$ is given by evolving the simple-growth SPGPE to equilibrium. Physically, the SPGPE describes the exchange of particles and energy between the c-field $\psi$ and a static thermal reservoir of chemical potential $\mu$ and temperature $T$. Consequently, it eventually evolves any initial state to thermal equilibrium in the grand canonical ensemble [41]. The value of the number-damping strength $\gamma$ does not have any impact on the equilibrium properties in SPGPE theory, and thus may be chosen to give rapid convergence to equilibrium. In our simulations $\phi_\text{therm}$, is sampled by evolving Eq. (2) with $\gamma_0 = 1.0$ for 100 trapping periods. Crucially, this sampling method provides the state of an *interacting* thermal Bose gas at equilibrium within the c-field approximation.

The combination of the noise $\eta$ and the stochastic noise in the thermal state sampling of $\phi_\text{therm}$ gives an initial simulation state that includes both quantum and thermal fluctuations in the grand canonical ensemble. Projecting the noise $\eta$ via the projector $\mathcal{P}\{*\}$ ensures that the c-field remains within the low-energy subspace **C**. In practice, the projector is implicitly included when adding the noise in the single-particle basis (analogous to the projection of the noise term in the SPGPE, which is described in Appendix A).

A consequence of including quantum fluctuations in our initial state is that there will be formal corrections in the calculation of observables from the c-field $\psi$, due to the non-commutativity of the quantum field operators $\hat{\psi}_\mathbf{C}$ with their conjugate $\hat{\psi}_\mathbf{C}^\dagger$ [41]. However, due to the large atom numbers studied in this work (on the order of $10^5$), these corrections are small and can thus be neglected.

---

[2]Explicitly, $|\phi_\text{therm}\rangle = \exp\left[\sqrt{N}(\hat{a}_{\phi_\text{therm}} - \hat{a}_{\phi_\text{therm}}^\dagger)\right]|\text{vac}\rangle$, where $\hat{a}_{\phi_\text{therm}} = \int d\mathbf{r}\, \phi_\text{therm}(\mathbf{r})\hat{\psi}_\mathbf{C}(\mathbf{r})$ and $N = \int d\mathbf{r}\, |\phi_\text{therm}(\mathbf{r})|^2$. Here $\phi_\text{therm}(\mathbf{r})$ is the position-space representation of $\phi_\text{therm}$.

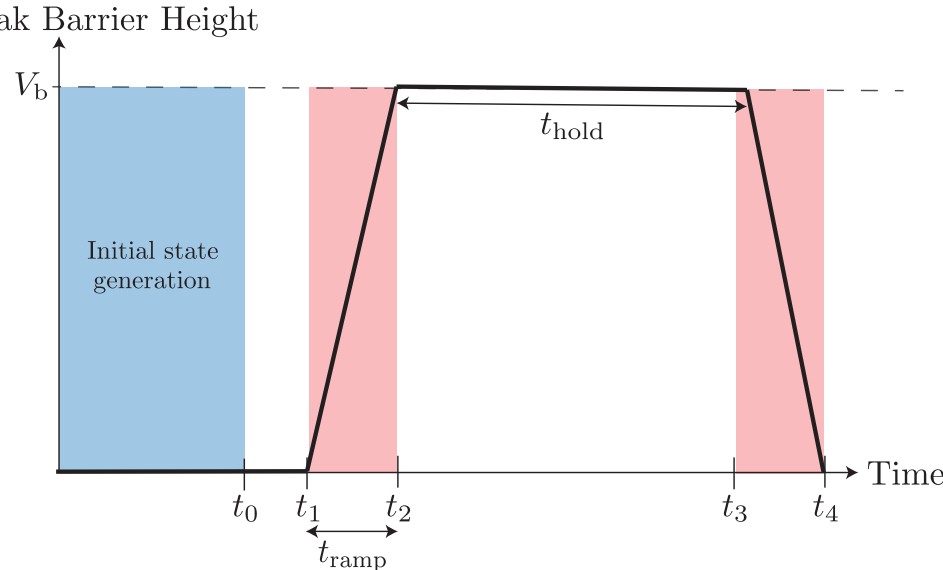

Figure 3: Schematic of the simulation protocol, characterised by peak barrier height $\max_{r,\theta}[U_b(r,\theta,t)] = \chi(t)V_b$ as a function of simulation time. In each simulation, the initial state is first generated (shaded blue) by evolving the SPGPE to equilibrium for $t < t_0$. At $t = t_0$, the initial state sample is seeded with quantum fluctuations and imprinted with a $2\pi$ phase winding. After some time $t_1 - t_0$, the barrier is linearly ramped up over a period of time $t_{\mathrm{ramp}}$ until it reaches its maximum height ($\chi = 1$) at $t = t_2$. The barrier is held at that maximum height for time $t_{\mathrm{hold}}$ and then linearly ramped down over the period $t_3 < t < t_4$.

### 3.4.1 Phase imprinting

In the experiment, an $l = 1$ circulation state is prepared by 'stirring' the perturbing barrier around the superfluid (see the Supplemental Material of Ref. [15]). Assuming this procedure perfectly prepares the metastable $l = 1$ circulation state, we may model this by instantaneously imprinting a $2\pi$ phase winding on our initial state:

$$\psi(0) \to \psi(0)e^{il\theta} \tag{13}$$

with $l = 1$. Modes with energy above the cutoff $\epsilon_{\mathrm{cut}}$ are incoherent, and so are unaffected by this transformation, and thus the thermal reservoir is treated as non-rotating.

### 3.5 Perturbing barrier and experimental sequence

In the experiment, the perturbing barrier is created by dithering a Gaussian beam in the radial direction such that its time average is [31]:

$$U_b(r,\theta,t) = \chi(t)\frac{V_b}{2}\left[\mathrm{erf}\left(\frac{\sqrt{2}}{w}(r - r_0 + l_d/2)\right) - \mathrm{erf}\left(\frac{\sqrt{2}}{w}(r - r_0 - l_d/2)\right)\right]e^{-\frac{2r^2(\theta - \theta_0)^2}{w^2}}, \tag{14}$$

where $w = 6\mu$m is the $1/e^2$ half-width of the Gaussian beam and $l_d = 21.8\mu$m is the width of the dither. The bracketed term ensures that the barrier vanishes at the edge of the torus. In our simulations, we choose the barrier to be centered at $\theta_0 = -\frac{\pi}{2}$. The time-dependent element $0 \le \chi(t) \le 1$ describes the raising and lowering ('ramping up/down') of the barrier.

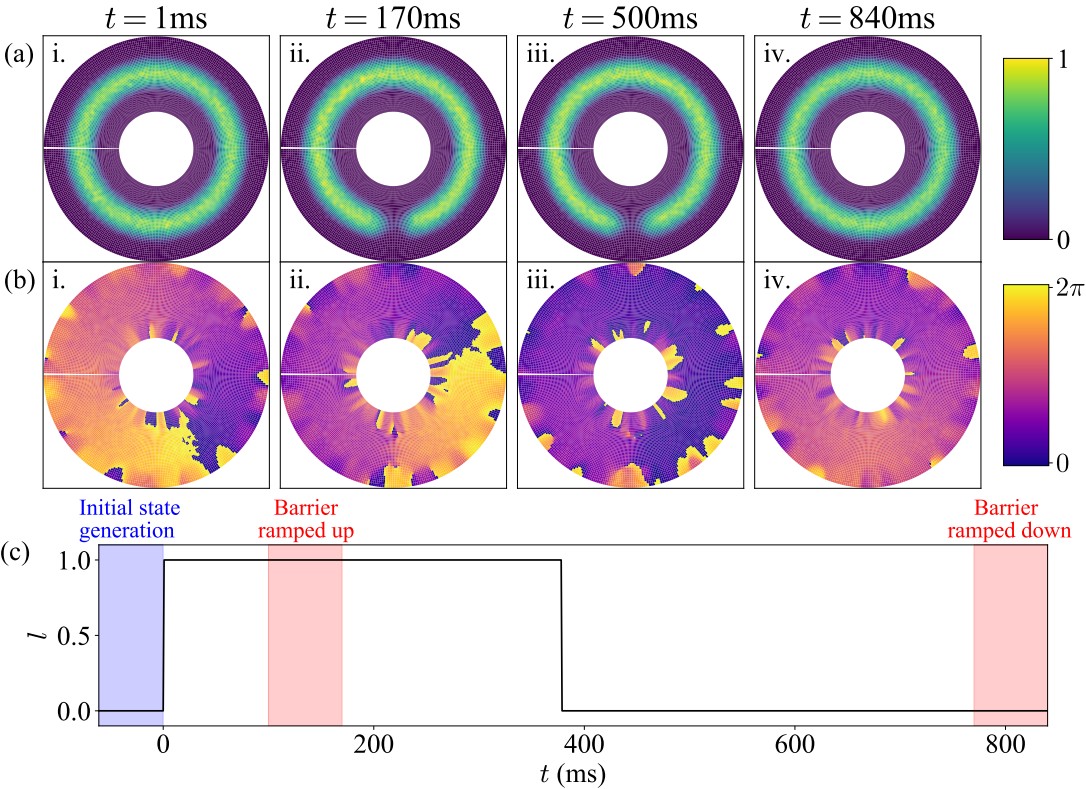

Figure 4: Single trajectory of the SPGPE for a simulation of the experimental sequence at $T = 85$nK with the barrier held at its maximum strength $V_b = 0.82\mu$ for $t_{\text{hold}} = 600$ms. Slices of the density normalised to 1 and phase in the $x - y$ plane are shown in (a) and (b), respectively, at different times. The winding number as a function of time is shown in (c), with the initial state generation period $t < 0$ (shaded blue) and barrier ramping periods (shaded red) shown. At time $t = 1$ms (i), the superfluid is in the metastable $l = 1$ circulation state, with a $2\pi$ winding of the phase profile. The barrier is raised in strength over the period $100$ms $< t < 170$ms. At $t = 170$ms (ii), the barrier reaches its maximum strength and the density in the region of the barrier is visibly depleted. The superflow decays stochastically from $l = 1$ to $l = 0$ during the period where the barrier is held at its maximum $170$ms $< t < 770$ms. After the decay event occurs (iii), the density remains depleted in the region of the barrier, however the $2\pi$ phase winding vanishes. The barrier is then ramped down over the period $770$ms $< t < 840$ms, at the end of which (iv) the density in the region of the barrier is restored.

The simulation protocol closely follows the experimental sequence and is shown in Figure 3. After the initial state is prepared as described in Sec. 3.4, the barrier is ramped up in strength over some time $t_{\text{ramp}}$, held at its maximum value ($\chi = 1$) for $t_{\text{hold}}$, and then ramped down over $t_{\text{ramp}}$. The ramp is linear, e.g. during the ramp-up sequence $\chi(t) = (t - t_1)/t_{\text{ramp}}$. Following the experiment, our simulations use $t_{\text{ramp}} = 70$ms.

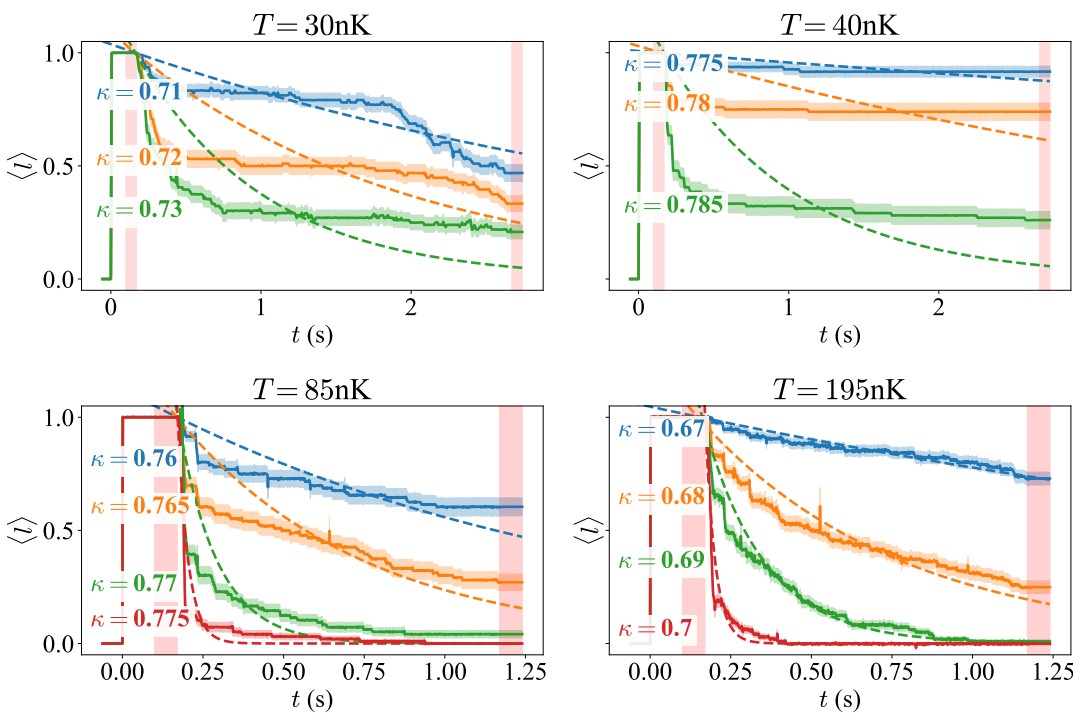

Figure 5: Average winding number as a function of time, for different temperatures and barrier heights, which are scaled by the chemical potential, $\kappa = V_b/\mu$. Shaded bars give a 95% confidence interval based on the standard error. The period of time $t < 0$ corresponds to initial state generation, with quantum fluctuations and $2\pi$ phase winding imprinted at $t = 0$. Vertical red shaded regions denote ramping-up/down of the perturbing barrier. For a given temperature, increasing the barrier height decreases the stability of the $l = 1$ circulation state, leading to faster decay of $\langle l \rangle$. Dashed lines are fits of the form $\langle l \rangle = c_0 \exp\left((t - t_2)/\tau\right)$, fitted to simulation data in the time period between the shaded red regions, where $t_2 = 0.17$s is the time taken for the barrier to reach its maximum height.

## 4 Results and Analysis

### 4.1 Qualitative features

To ensure the validity of our numerical procedure, we compare qualitative features of our simulations to the experiment of Ref. [15], as well as to the theoretical work of Mathey *et al.* [32], who attempted to model the related experiment of Ramanathan *et al.* [10].

As noted by Mathey *et al.*, zero-temperature mean-field theory (the GPE) predicts there is either no decay over the lifetime of the experiment or rapid decay (in less than 10ms), depending on whether the value of $V_b$ is above some critical value or not. This suggests the dynamics of the superflow is driven by Hamiltonian dynamics well captured by mean field theory, for very small and very large barrier heights. This behaviour is to be contrasted with the superflow decay observed both in the experiment of Kumar *et al.* [15] and the finite-temperature simulations conducted by Mathey *et al.* [32], where stochastic decay events were observed at both short and long time scales, specifically across the entire time period where the barrier was held at its maximum. We observe qualitatively similar stochastic decay events in our simulations. As an example, the results from a single trajectory of the SPGPE are shown in Fig. 4, in which a decay event occurs roughly 110ms

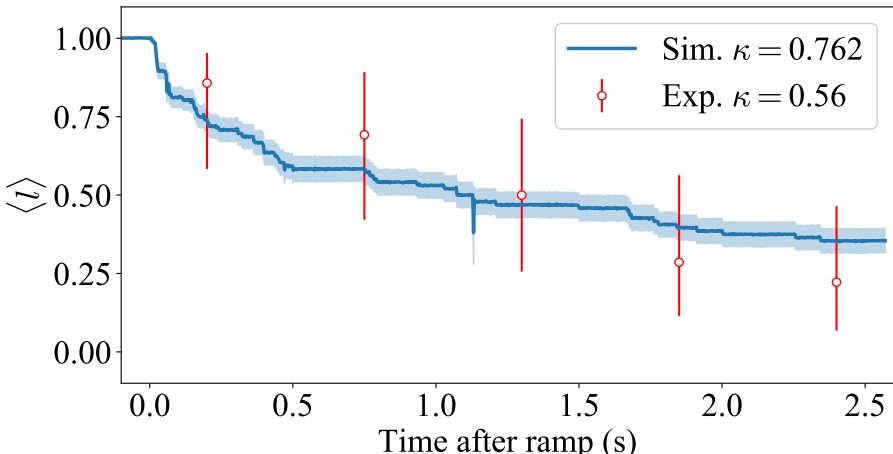

Figure 6: Average winding number as a function of time after the barrier reaches its maximum height $(t - t_2)$ for $T = 85\text{nK}$. The normalised barrier height $\kappa \equiv V_b/\mu = 0.762$ for the simulation result (blue) is chosen to give close agreement with the experimental data for $\kappa = 0.56$ (red open circles). The discrepancy in the barrier heights is quantified in Section 4.2 and discussed further in Sec. 5. Shaded region and errors bars give a 95% confidence interval in the simulation and experimental data, respectively.

after the barrier reaches its maximum height.

In Fig. 5 we present the results of SPGPE simulations for the four different temperatures $T = 30, 40, 85, 195\text{nK}$ studied in the experiment[3]. For reference, the critical temperature is $T_c = 370\text{nK}$ for $T = 40, 85\text{nK}$ and $T_c = 470\text{nK}$ for $T = 30, 195\text{nK}$, as given in the Supplemental Material of Ref. [15]. For each of the four temperatures, simulations are run for several different maximum barrier heights within a range of no more than $\sim0.05\mu$. In any given trajectory, the transition of the winding number from $l = 1$ to $l = 0$ occurs stochastically, with the average value decaying slowly over a range of timescales $\sim100\text{ms}-10\text{s}$. In Fig. 6, we show that the decay of the average winding number for $T = 85\text{nK}$ is similar to the experimental results of Kumar *et al.*, albeit for different values of barrier strength $V_b$ (this will be discussed further in the following section).

In both the experiment of Kumar *et al.* and the simulations by Mathey *et al.*, the decay of the superflow while the barrier is at maximum strength appear well-fitted by decaying exponential trends. In Fig. 5 we can see that the results of SPGPE simulations for the lower temperatures studied do not seem to be well-suited to an exponential fit. While an exponential decay is not necessarily expected in the presence of nonlinearities, this behaviour may also be an artefact of ensemble averaging over a relatively small number of stochastic trajectories (96), or it might be suggestive of approximations made in the derivation of the SPGPE breaking down at low temperatures and long simulation times. The latter is discussed in more detail in Sec. 5.1. However, fitting these non-exponential trends to a simple exponential is still useful in estimating the timescale of the decay. In Appendix C we show that the trends are better captured by a two-timescale fit. However, it is not clear whether these 'timescales' are physically motivated, and thus is difficult to compare them to the experimental results. Furthermore, the dominant timescales calculated within this approach are not significantly different to the single-timescale model, and thus there is no apparent advantage in using a two-timescale model in this study.

---

[3]Note that each temperature is associated with different experimental parameters (specifically, the trapping frequencies and atom number change with temperature), and thus results for different temperatures should not be naïvely compared with each other.

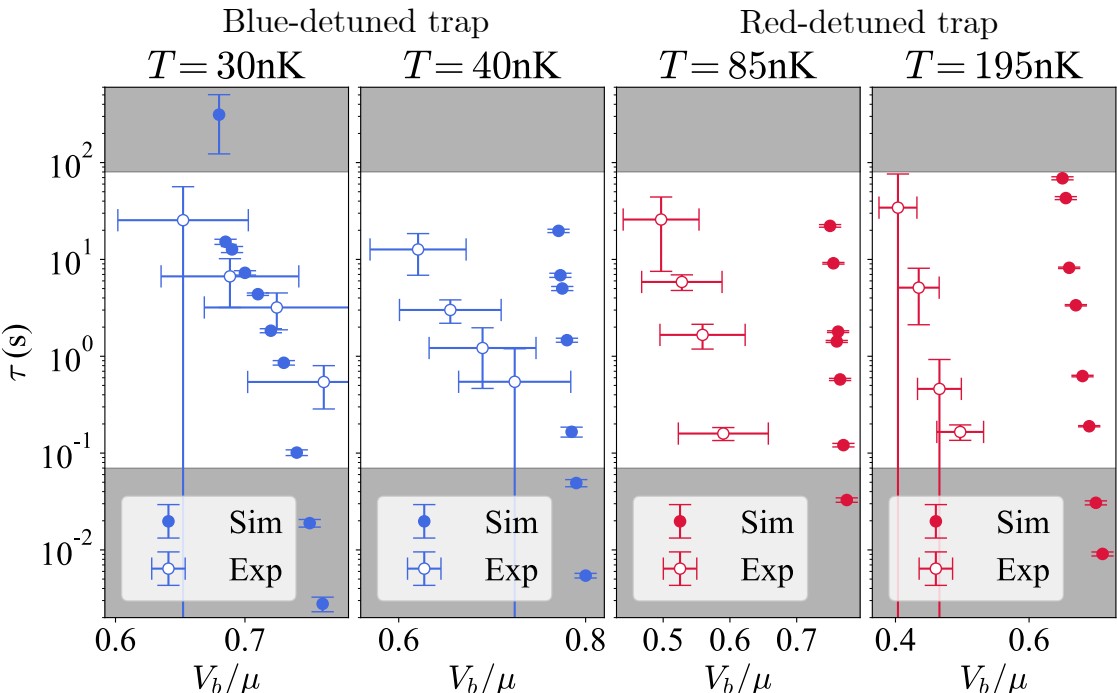

Figure 7: Comparison of the superflow lifetime $\tau$ between simulations (closed circles) and experiment (open circles) as a function of normalised barrier height $V_b/\mu$, for the four different temperatures studied in the experiment. Error bars on the experimental data give a 95% confidence interval. The grey shaded areas denote the timescales that sit outside the estimated range of values detectable by the experiment. The simulation and experimental data show strong quantitative agreement for $T = 30$nK, however there is an increasing discrepancy at higher temperatures. Specifically, the simulations suggest a larger barrier height is required to model a given decay time, and that this 'offset' increases with temperature.

## 4.2 Quantitative comparison to experiment

As described in the section above, the timescale of superflow decay $\tau$ can be quantified by fitting the average winding number to the trend

$$\langle l \rangle = c_0 \exp\left(-\frac{(t - t_2)}{\tau}\right) \qquad (15)$$

where $c_0$ and $\tau$ are positive-valued fitting parameters[4]. The fit is over the window of time $t_2 \leq t \leq t_3$ (see Fig 3), which is the period where the barrier is held at its maximum height. Simulations are performed for a range of barrier heights at each temperature, some of which for $t_{\text{hold}} = 2.5$s and the remainder for $t_{\text{hold}} = 1$s (depending on whether there was sufficient decay to reasonably fit). The simulation time ultimately places bounds on the range of timescales $\tau$ accessible in the simulations.

The comparison of the computed timescale $\tau$ to that measured in the experiment is shown in Fig. 7, for each of the four temperatures. We can see that the timescale changes by four orders of magnitude over a small range of normalised barrier heights ($V_b/\mu$), with roughly exponential trends (linear trends in log-linear space). In particular, the simulation results for $T = 30$nK closely agree with the experiment. For higher temperatures, there is

---

[4]For all the fits performed in this section, $c_0 \approx 1$. To ensure the best possible fit we do not fix $c_0 = 1$, although in practice this has a negligble effect on our results.

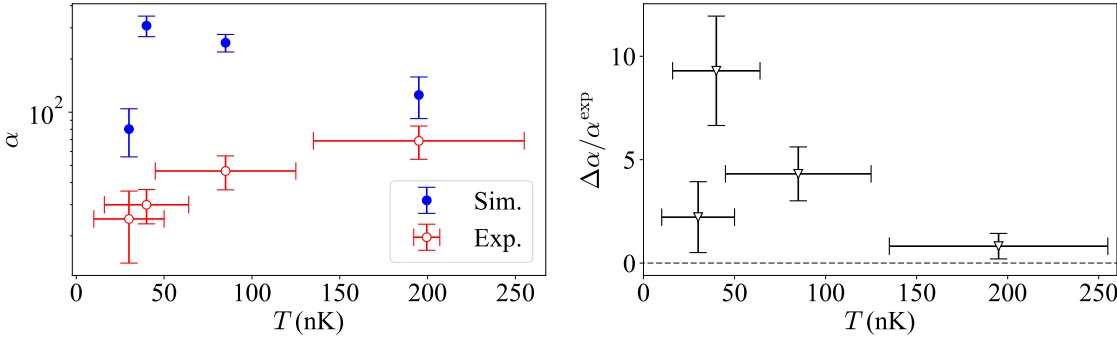

Figure 8: The sensitivity of the decay timescale $\tau$ to small changes in barrier height, characterised by the fit parameter $\alpha$, as a function of temperature. (Left) Comparison between the value calculated from the simulations $\alpha^{\mathrm{sim}}$ (blue filled circles) and experimental data $\alpha^{\mathrm{exp}}$ (red open circles). While the experimental results show the sensitivity monotonically increasing with temperature, the simulation trend qualitatively differs. (Right) The relative discrepancy between the simulation and the experimental values, $\Delta\alpha = \alpha^{\mathrm{sim}} - \alpha^{\mathrm{exp}}$. The dashed horizontal line illustrates $\Delta\alpha = 0$. The discrepancy is greatest for $T = 85\mathrm{nK}$ and least for the highest temperature $T = 195\mathrm{nK}$. Error bars give a 95% confidence interval.

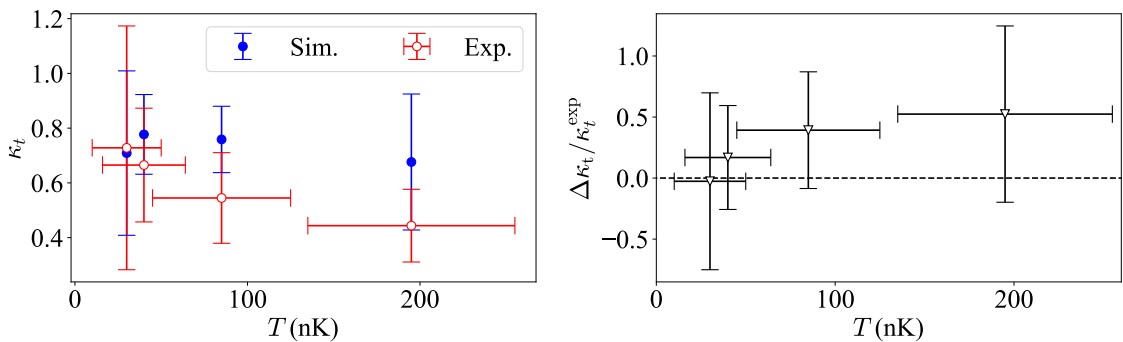

Figure 9: The normalised barrier height for which the decay timescale is predicted to be $\tau = 1\mathrm{s}$, $\kappa_t$, as a function of temperature. (Left) Comparison between the experimental (red open circles) and simulation values (blue filled circles). The experimental trend is monotonically decreasing in temperature, which is not qualitatively reflected in the simulation trend. (Right) The relative discrepancy between the simulation and the experimental values, where $\Delta\kappa_t = \kappa_t^{\mathrm{sim}} - \kappa_t^{\mathrm{exp}}$. The dashed horizontal line illustrates $\Delta\kappa_t = 0$. The error between the experimental and simulation values increases monotonically with temperature. Error bars give a 95% confidence interval.

greater discrepancy between the experiment and simulation results, although the trends remain qualitatively similar. For instance, there is a discrepancy in the sensitivity of $\tau$ to changes in the barrier height (i.e. the slope of the trends in Fig. 7); this discrepancy is largest for the intermediate temperature $T = 40\mathrm{nK}$ and $T = 85\mathrm{nK}$. More significantly, the trends in Fig. 7 suggest that our model requires a larger barrier height to reproduce superflow decay with a given timescale $\tau$. This 'offset' in the barrier height is largest for the highest temperature, $T = 195\mathrm{nK}$.

To quantify these disparities, we fit the trends in Fig. 7 to the following exponential

function:

$$\tau = a \exp\left(-\alpha \frac{V_b}{\mu}\right) . \tag{16}$$

The dimensionless fit parameter $\alpha$ captures the sensitivity of the decay timescale to changes in the barrier height. As shown in Fig. 8, the values of $\alpha$ for the simulation results are larger than the experimental results for all temperatures. Furthermore, the trends are qualitatively different; the experimentally-estimated values of $\alpha$ increase monotonically with temperature, which is not reflected in the simulation results. The quantitative discrepancy is greatest for $T = 40$nK, with simulations overestimating $\alpha$ by an order of magnitude. Agreement is strongest for the highest temperature studied $T = 195$nK, with the simulation value lying within 100% of the experimental value. This may be in part due to the exponential function being a poor fit to the simulation data for the lower temperatures (c.f. Fig. 5).

To quantify the discrepancy in the magnitude of the barrier height at which superflow decay begins to occur (i.e. the 'offset' seen in Fig. 7), we introduce the quantity $\kappa_t$, which is the normalised barrier height for which the model predicts a one second superflow lifetime. This quantity is estimated by inverting Eq. (16) to get $\kappa_t = V_b/\mu$ for $\tau = 1$s. In Fig. 9 we can see that the discrepancy between the value obtained from the simulations and experimental data $\Delta\kappa_t$ increases monotonically with temperature, with strong agreement for the lowest temperature $T = 30$nK.

## 5 Discussion

At each temperature studied, save perhaps for the lowest temperature $T = 30$nK, we have observed significant discrepancy between the predictions of our model and the experimental results of Ref. [15]. For the two intermediate temperatures studied, $T = 40, 85$nK, the observed discrepancies could feasibly be due to the inadequacy of our model in the lower temperature limit (see Sec. 5.1.1). However, for the lowest and highest temperatures studied, we expect our model to give quantitative agreement with the experimental results, as the validity of zero-temperature TW and SPGPE theory has been well-demonstrated at low and high temperatures, respectively. This suggests that the large discrepancy observed between the results of our simulations and the experimental data for the largest temperature, $T = 195$nK, is likely due to additional experimental features not captured by our model.

In this section we will detail possible sources for the discrepancy, including assumptions and limitations of our theoretical model as well as technical effects within the experiment.

### 5.1 Limitations of the theoretical model

#### 5.1.1 Validity in the low-temperature limit

As SPGPE theory is formulated in the high-temperature regime $(\sim T_c/2 - 1.1 T_c)$ where there is a significant thermal fraction of the gas, it is not *a priori* valid in the low-temperature regime of the experiment. Specifically, within the derivation of the SPGPE, reservoir interactions are expanded in powers of $(\mu - \mathcal{L})/k_{\mathrm{B}}T$, and truncated at low order, the validity of which is well satisfied by the requirement $k_{\mathrm{B}}T \gg \mu$. The truncation may be valid for much lower temperatures if the system is not far from particle equilibrium with the reservoir. Since the experiment does not operate in the high-temperature regime (for all temperatures except $T = 195$nK, $k_{\mathrm{B}}T < \mu$) it is possible this truncation discards

important reservoir interactions for the two intermediate temperatures $T = 40, 85$nK studied in this work. In particular, omission of these higher-order reservoir interactions may be the source of the discrepancy in the sensitivity of the decay timescale on barrier height, as shown in Fig. 8. In addition, it may also account for the non-exponential nature of the trends for the lowest temperatures ($T = 30, 40$nK) in Fig. 5. Extensions of the SPGPE which include the higher-order terms will likely be challenging to implement numerically, however may be a useful avenue of future work.

### 5.1.2 Truncated Wigner Approximation

The SPGPE is derived by mapping a high-temperature master equation for the **C** region's quantum state to a partial differential equation (PDE) for the state's Wigner function. In general, this PDE suffers from the same computational intractability as the original master equation. However, by making the truncated Wigner approximation, which neglects third- and higher-order derivatives in this PDE, the resulting equation of motion for the Wigner function takes the form of a Fokker-Planck equation that can be efficiently simulated via the SPGPE. The validity of our SPGPE simulations therefore depends in part upon the validity of the truncated Wigner approximation.

Although the truncated Wigner approximation is an uncontrolled approximation[5], its validity has been verified in the classical field regime where the particle occupation per mode is high [41, 88, 89]. This condition is easily fulfilled in our work by the inclusion of a projector and our choice of energy cutoff. Nevertheless, the truncated Wigner approximation is only valid for a finite simulation time, since the (unquantifiable) error in the truncation will compound with time [48]. The simulations performed in this work have been over a long timescale, relative to typical cold-atom experiments, and thus it is feasible that the truncated Wigner approximation may begin to appreciably breakdown for the longest simulations ($t_{\text{hold}}$=2.5s). This effect is stronger in the low-temperature regime where the density of the atomic cloud is higher and there is more scattering, which may account for the qualitative trends in Fig. 5 for $T = 30$nK, which deviate significantly from exponential trends after about $t \approx 2$s.

### 5.1.3 Static thermal reservoir

Our SPGPE model assumes that the thermal reservoir (high-energy modes above $\epsilon_{\text{cut}}$) is static and unaffected by the dynamics of the **C** region. The typical justification for this is that the sparsely-populated high-energy modes equilibrate rapidly compared to the macroscopically-occupied modes of the **C** region. While there are many systems for which the dynamics of the thermal cloud are an essential aspect of the physics (for example, collective modes of the condensate and thermal cloud [90]), there is currently no formulated extension of SPGPE theory that includes the dynamics of the **I** region. To study these effects, an alternative theoretical framework must be used, such as the coupled condensate-thermal theory of Zaremba, Nikuni, and Griffin [91]. However, it is incorrect to say the SPGPE does not treat the thermal cloud dynamically. The majority of thermal atoms are contained within the degenerate modes of the **C** region; it is only the high-energy sparsely-populated thermal modes that make up the thermal reservoir. In this sense, the majority of the thermal cloud is treated on the same footing as the condensate

---

[5]It has been argued that the truncated Wigner approximation is a controlled approximation since, in principle, it is possible to calculate higher-order corrections to scattering processes neglected by the truncated Wigner approximation [87]. In practice, this is unachievable for large multimode calculations, such as those undertaken in this work. Consequently, for all practical purposes, the truncated Wigner approximation is best considered an uncontrolled approximation.

in SPGPE theory. Given the small thermal fractions considered in this work, most of which are contained within the **C** region, it is unlikely that neglecting non-equilibrium dynamics of the **I** region significantly affects the simulated decay timescales.

Despite these small thermal fractions, the atoms in the **I** region cannot simply be neglected. This is the approach adopted when using the projected GPE (PGPE), a c-field theory that only describes the portion of the thermal cloud within the **C** region [92, 93]. A key attraction to this approach is its computational simplicity relative to the simple-growth SPGPE; its lack of dissipative and dynamical noise terms make it less challenging to numerically integrate than the SPGPE. In situations where the thermal cloud is small and the system is near equilibrium, the finite-temperature PGPE has been able to quantitatively describe experiments both in and out of equilibrium [94, 95]. Within the context of this work, the PGPE captures the qualitative nature of the stochastic decay observed in the experiment of Kumar *et al.* [15]. However, quantitatively it provides a poorer description of the experiment than our SPGPE model, due to the lower rate of dissipation relative to the simple-growth SPGPE (see Fig. 18 in Appendix D). This, alongside the strong temperature-dependence of the experimentally-observed superflow decay, suggests that it is important to retain dissipation due to scattering processes with atoms in the **I** region. Indeed, *increasing* this dissipation via increasing $\gamma$ could potentially improve the agreement between our model and the experiment, although there is no *a priori* justification for increasing $\gamma$ in this way.

### 5.1.4 Parameter estimation

In this work, the SPGPE parameters that describe reservoir interactions are estimated self-consistently using the experimentally-measured atom numbers and temperatures. Nevertheless, to account for possible uncharacterised errors in these measurements, we investigated the sensitivity of our simulation results to changes in several parameters, including barrier width and temperature. Changes to these parameters do not strongly change our results, suggesting that the imprecise experimental calibration of these parameters cannot account for the discrepancy between our simulations and experimental data. We also varied the energy cutoff, which is not an experimental parameter, and confirmed that our results were not strongly cutoff dependent. Appendix B explains how we estimated our parameter values and fully details the effect of parameter variations on our simulation results.

## 5.2 Uncharacterised experimental effects

### 5.2.1 Shot-to-shot number fluctuations

Shot-to-shot fluctuations in the total atom number is one example of a common technical effect in experiments with ultracold atomic gases, the magnitude of which can certainly be temperature dependant. In fact, fluctuations in atom number were proposed as a possible cause of the discrepancy between experimental results in a related experiment (see the Supplemental Material of Ref. [39]). The simulations performed in this work include some fluctuations in the atom number, as the initial thermal state is sampled from the grand canonical ensemble. However, these fluctuations are no more than 3% for $T = 195$nK, and less than 1% for $T = 30$nK. We can increase the magnitude of shot-to-shot atom number fluctuations in our simulations by sampling the chemical potential $\mu$ from a Gaussian distribution with mean $\bar{\mu}$ and width chosen to give a certain variance in the total atom number. As an example, we study the inclusion of atom number fluctuations for a fixed barrier height $V_b/\bar{\mu} = 0.78$ and $T = 40$nK, in Fig. 10. We find that the average winding

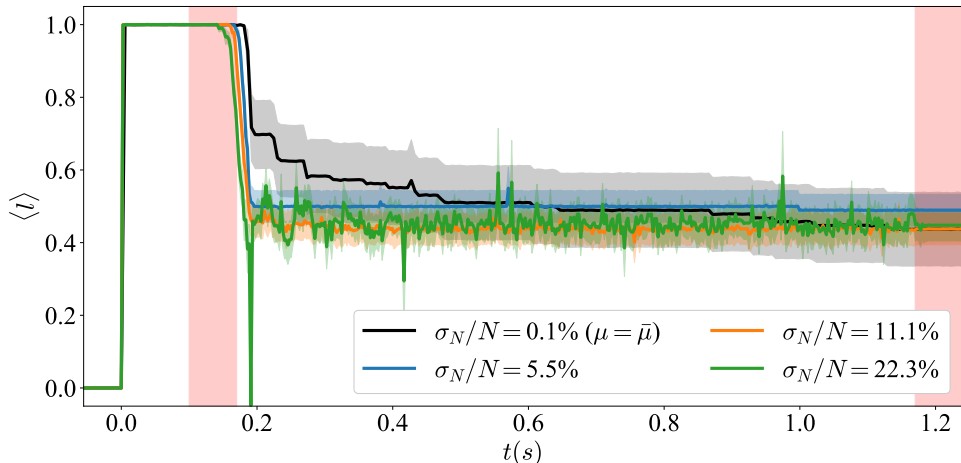

Figure 10: The effect of shot-to-shot fluctuations in the total atom number, for a barrier height of $V_b = 0.78\mu$ and a temperature of $T = 40$nK. When the chemical potential is kept constant from shot-to-shot, $\mu = \bar{\mu}$, there are fluctuations in the atom number on the order of 0.1%, due to sampling the initial state from the grand canonical ensemble. Larger shot-to-shot atom number fluctuations are modelled by randomly sampling the chemical potential $\mu$ from a Gaussian distribution with mean $\bar{\mu}$. As the fluctuations increase in magnitude, the average winding number begins to decay earlier in time. However, the overall timescale for decay is not significantly affected.

number is not significantly affected for shot-to-shot atom number fluctuations of magnitude $5-20\%$, which is the size observed in typical ultracold atom experiments, suggesting that such fluctuations cannot account for the noted discrepancies.

### 5.2.2 Barrier Calibration

As shown in Fig. 9, the discrepancy between the experimental data and our simulations in $\kappa_t$ (the estimated value of $\kappa = V_b/\mu$ that gives $\tau = 1$s) increases with the temperature of the Bose gas. This could be caused by a temperature-dependent systematic effect in the barrier calibration, which is certainly possible as the experimental calibration of the normalised barrier strength $V_b/\mu$ did not differentiate between condensate and thermal atoms. The magnitude of this error may be estimated by computing the contribution of the thermal atoms to the chemical potential $\mu$ within a semiclassical approximation. This calculation is included in the Supplemental Material of Ref. [15], where the systematic shift in the barrier calibration was found to be approximately 3% for $T = 85$nK and 8% for $T = 195$nK. Not only is this smaller than the statistical error in the experimental calibration, but it is far smaller than the discrepancy in the range of barrier heights for which decay occurs between the experimental data and the simulation results, which is roughly 35% and 50% for $T = 85$nK and $T = 195$nK, respectively (see Fig. 9). Therefore, an unaccounted shift in the barrier calibration due to the presence of a thermal cloud cannot account for the discrepancies between the experimental results and the predictions of our model.

## 5.3   Optical trap imperfections

In general, technical noise on the trap lasers causes heating of an optically trapped super-fluid, which may lead to enhanced dissipation of superflow. Furthermore, in this experiment the superflow decay rate is particularly sensitive to small changes in barrier height. This could amplify the effect of slight violations of the trap's azimuthal symmetry in the experiment [96]. This breaks the symmetry of the ideal ring considered in the simulations, allowing more pathways for vortex escape, thus enhancing the rate of superflow decay. This may have contributed to the observation of superflow decay in the experiment at lower barrier heights than predicted by our model.

The experiments performed at higher temperatures used different optical trapping beams to the experiments performed at lower temperatures. As described in Sec. 2, the low temperature $T = 30, 40$nK BECs were confined in a blue-detuned dipole trap, whereas the higher temperature $T = 85, 195$nK BECs were confined in a red-detuned dipole trap. The red-detuned beams used in the experiment suffered from etaloning in the vacuum cell, which resulted in fringes in the trapping potential [96, 97]. The differences between the blue-detuned and red-detuned traps could account for the perceived temperature dependence of superflow decay in the experiment not modelled in our simulations.

## 6   Conclusions

In this work, we have performed detailed three-dimensional classical-field simulations to model the experiment of Kumar *et al.* [15]. Our model, which combines zero-temperature TW methodology and SPGPE theory, describes the role of quantum and thermal fluctuations in the spontaneous decay of persistent currents of a superfluid BEC trapped in a toroidal geometry. We have demonstrated that our model is able to capture the essential non-equilibrium dynamics that lead to superflow decay in the presence of a perturbing barrier, in good qualitative agreement with the experimental results of Kumar *et al.* and a previous theoretical analysis of a related experiment [32]. Specifically, the predictions of our model are in quantitative agreement with the experiment at low temperature, and provide a qualitative description of the experiment at higher temperatures. Furthermore, our simulations predict the same range of decay timescales as observed in the experiment, across all temperatures studied. Notably, this is achieved with simulation parameters estimated solely from the experimental temperature and atom number.

For the lowest temperature Bose gas studied in the experiment, the decay timescales predicted by our model quantitatively agree with the experimental data. For the other temperatures studied, however, we have found discrepancies between the quantitative predictions of our model and the experimental data, which become most significant at the highest temperature. As discussed in Section 5.2, this is likely not solely due to limitations of the theoretical model, as there were several technical effects in the experiment that may have lead to enhanced superflow decay.

In general, obtaining quantitative agreement with experimentally measured decay rates is very difficult to achieve due to the many sources of dissipation in a real superfluid. In particular, modelling the experiment of Ref. [15] was a challenging task, as a three-dimensional model was required that included the effects of both quantum and thermal fluctuations. Moreover, we have pushed the SPGPE model beyond its safe regime of applicability for studying the dynamics of a Bose gas in the low-temperature regime of the experiment, and noted a range of technical effects that could lead to enhanced dissipation of the superflow. Despite this, we have observed some level of quantitative agreement without fitted parameters for low temperatures, providing further evidence on the adequacy of a

c-field description of highly non-equilibrium dynamics in Bose gases. Nevertheless, our work suggests a need for further theory beyond SPGPE, alongside further experimental characterisation of superflow decay at higher temperatures.

## Acknowledgements

We thank A. Kumar and G. K. Campbell for providing experimental data and for insightful comments on the experimental setup. We acknowledge useful conversations with Y. Ben Aicha, S. A. Haine, and R. J. Thomas. This research was undertaken with the assistance of resources and services from the National Computational Infrastructure (NCI), which is supported by the Australian Government.

**Funding information**   ZM is supported by an Australian Government Research Training Program (RTP) Scholarship. ASB acknowledges financial support from the Marsden Fund (Grant No. UOO1726) and the Dodd-Walls Centre for Photonic and Quantum Technologies. SSS is supported by an Australian Research Council Discovery Early Career Researcher Award (DECRA), Project No. DE200100495.

## A   Numerical methods

Within classical-field methodology, the numerical implementation of a well-defined energy cutoff is crucial in order to make quantitative physical predictions [41]. This can be achieved by projecting the equation of motion for the classical field onto a basis where the many-body Hamiltonian is approximately diagonal. At the high energies where the cutoff is typically imposed, this is satisfied by the eigenbasis of the single-particle Hamiltonian $H_0$. This corresponds to the decomposition of the classical field as:

$$\psi(r, \theta, z) = \sum_{\mathbf{n} \in \mathbf{C}} \psi_{\alpha n \Gamma} \Phi_{\alpha n \Gamma}(r, \theta, z) \,, \tag{17}$$

where $\Phi_{\alpha n \Gamma}(r, \theta, z)$ are the single-particle modes of $H_0$.

In the context of toroidal confinement, the eigenstates of the potential in Eq. (5) are not analytically known. However, recent work by Prikhodko *et al.* [84] has demonstrated that an *approximate* single-particle basis may be used:

$$\Phi_{\alpha n \Gamma}(r, \theta, z) = \frac{1}{\sqrt{2\pi r}} \varphi_\alpha^{(\omega_r)}(r - r_0) e^{in\theta} \varphi_\Gamma^{(\omega_z)}(z) \,, \tag{18}$$

where $\varphi_\alpha$ are the normalised Hermite-Gauss functions (in physical units):

$$\varphi_\alpha^{(\omega)}(x) = \frac{1}{\sqrt{2^\alpha \alpha!}} \left( \frac{M\omega}{\pi\hbar} \right)^{1/4} \exp\left( -\frac{M\omega x^2}{2\hbar} \right) H_\alpha\left( \sqrt{\frac{M\omega}{\hbar}} x \right) \tag{19}$$

using the physicists' Hermite polynomials $H_n(x)$. As described in Ref. [84], this basis is approximately orthonormal provided it is truncated such that the highest-energy mode vanishes as $r \to 0$.

In this limit, this basis diagonalises the single-particle Hamiltonian with the energy

spectrum

$$E_\alpha^{(r)} = \hbar\omega_r(\alpha + \frac{1}{2}) \, , \tag{20}$$

$$E_\Gamma^{(z)} = \hbar\omega_z(\Gamma + \frac{1}{2}) \, , \tag{21}$$

$$E_n^{(\theta)} = \frac{\hbar^2}{2mr_0^2}\left(n^2 - \frac{1}{4}\right) \, . \tag{22}$$

This allows us to define the low-energy region **C** by only including modes with energies below the cutoff:

$$\mathbf{C} = \left\{ (\alpha, n, \Gamma) : \hbar\omega_r\left(\alpha + \frac{1}{2}\right) + \hbar\omega_z\left(\Gamma + \frac{1}{2}\right) + \frac{\hbar^2}{2mr_0^2}\left(n^2 - \frac{1}{4}\right) \leq \epsilon_{\text{cut}} \right\} \, . \tag{23}$$

Numerically, the projector is implemented by setting the occupations of single-particle modes with energies above $\epsilon_{\text{cut}}$ to zero.

Casting the projected GPE (Eq. (2) with $\gamma = 0$) onto this single-particle basis is detailed explicitly in Ref. [84], so we will not describe it here. Implementing the number-damping reservoir interaction terms in this basis is straightforward extension of their method. For the deterministic terms only the constant prefactors need to be adjusted. The number-damping noise term can be directly sampled on the single-particle basis via:

$$\mathcal{P}\{d\xi_\gamma(\mathbf{r})\} = \sqrt{\frac{2\gamma k_{\text{B}}T}{\hbar}} \sum_{\alpha, n, \Gamma \in \mathbf{C}} \Phi_{\alpha n \Gamma}(r, \theta, z) dW_{\alpha n \Gamma} \, , \tag{24}$$

where $dW_{\alpha n \Gamma}$ is a complex Weiner noise satisfying

$$\mathbb{E}[dW_{\alpha' n' \Gamma'}^* dW_{\alpha n \Gamma}] = \delta_{\alpha'\alpha}\delta_{n'n}\delta_{\Gamma'\Gamma}dt \, . \tag{25}$$

Similarly, the quantum fluctuations seeded in the initial states (Section 3.4) can also be directly sampled on the single-particle basis:

$$\frac{1}{\sqrt{2}}\mathcal{P}\{\eta(\mathbf{r})\} = \frac{1}{\sqrt{2}} \sum_{\alpha, n, \Gamma \in \mathbf{C}} \Phi_{\alpha n \Gamma}(r, \theta, z)\eta_{\alpha n \Gamma} \tag{26}$$

where $\eta_{\alpha n \Gamma}$ is a complex Gaussian noise satisfying

$$\mathbb{E}[\eta_{\alpha' n' \Gamma'}^* \eta_{\alpha n \Gamma}] = \delta_{\alpha'\alpha}\delta_{n'n}\delta_{\Gamma'\Gamma} \, . \tag{27}$$

We numerically integrate the SPGPE using the open source XMDS2 software package [98], exploiting an adaptive fourth-fifth order Runge-Kutta algorithm. The use of a high-order adaptive Runge-Kutta algorithm is appropriate due to the additive nature of the noise term in the simple-growth SPGPE, as first noted in Appendix B of Ref. [62]. In all the simulations reported in this work, the relative error tolerance of the algorithm is set at $10^{-5}$.

Transformations between the single-particle basis and spatial grids (where the nonlinear $|\psi|^2\psi$ term is diagonal) are implemented using in-built Hermite-Gauss and Fourier transforms in XMDS. Due to the presence of the $1/\sqrt{r}$ in the single-particle basis - Eq. (18) - Hermite-Gauss quadrature methods are inexact for computing spatial integrals with a finite number of grid points. To minimise this as a source of numerical error, we include an additional 16 points on the spatial quadrature grids.

| $T$ (nK) | $k_B T/\hbar\omega_r$ | $\omega_z/2\pi$ (Hz) | $\mu/\hbar\omega_r$ | $\epsilon_{\text{cut}}/\hbar\omega_r$ | $\gamma$ | $n_r \times n_\theta \times n_z$ |
|---|---|---|---|---|---|---|
| 30 | 2.42175 | 974 | 12.28 | 13.9586 | $3.4322 \times 10^{-6}$ | $12 \times 150 \times 4$ |
| 40 | 3.229 | 518 | 10.66 | 12.8982 | $4.6116 \times 10^{-6}$ | $12 \times 150 \times 6$ |
| 85 | 6.86162 | 520 | 10.3 | 15.0561 | $1.0282 \times 10^{-5}$ | $14 \times 150 \times 7$ |
| 195 | 15.7414 | 985 | 11.66 | 22.5711 | $2.5289 \times 10^{-5}$ | $21 \times 150 \times 6$ |

Table 1: Fixed parameters for the simulations performed in this work. Here $n_r, n_\theta, n_z$ are the number of single-particle modes in the $r, \theta, z$ dimensions, respectively.

## B  Fixing simulation parameters

The estimated simulation parameters for each of the four temperatures studied in this work are given in Table 1.

### B.1  Chemical potential

In our simulations we choose the value of the chemical potential $\mu$ such that the initial thermal states have an average atom number as close as possible to the experimentally reported value, for each temperature. Although this can be determined through a semi-classical calculation (see Appendix A of Ref. [63]), we find it convenient to simply vary $\mu$ until we achieve the desired atom number. Specifically, we first estimate the chemical potential by assuming a purely Thomas-Fermi density:

$$\mu = \hbar\sqrt{\omega_r\omega_z}\sqrt{\frac{2Na_s}{\pi r_0}}\ . \tag{28}$$

We then calculate the atom number at thermal equilibrium for a range of $\mu$ around this initial estimate, and fit the resulting trend to determine the value of $\mu$ that will give an atom number closest to the experimental value. Once the chemical potential has been set, the energy cutoff $\epsilon_{\text{cut}}$ and number-damping strength $\gamma$ can be estimated using Eqs. (6) and (7) in the main text.

### B.2  Energy cutoff

As described in Sec. 3.2, the energy cutoff in Eq. (6) is chosen to give an average occupation of $\overline{n} \approx 1$ for the single-particle modes near the energy cutoff, as is typically done in SPGPE analyses. We check this is satisfied, for simulations of the experimental sequence at $T = 40\text{nK}$ and $T = 195\text{nK}$, by computing the occupation of each single-particle mode $|\psi_{\alpha n\Gamma}|^2$ and plotting it as a function of its energy. This is shown in Fig. 11, where it is clear that the modes near the cutoff have an average occupation on the order of $\overline{n} \approx 1$.

As a more comprehensive check of the energy cutoff, in Fig. 12 we assess the quantitative impact of changing the cutoff value slightly around its estimated value, for the temperature $T = 85\text{nK}$ and a barrier height of $V_b = 0.77\mu$. Although the precise value of $\langle l \rangle$ changes slightly as $\epsilon_{\text{cut}}$ is varied, the decay timescale is not significantly affected. Specifically, increasing the energy cutoff by 20.0% leads to $\tau$ reducing to roughly 45% of its value at the estimated cutoff. Given that changing the barrier height by $\sim 0.1\mu$ changes $\tau$ by several orders of magnitude, this variation with $\epsilon_{\text{cut}}$ is acceptably small.

### B.3  Grid size

The size of the single-particle grid used in the simulations is set by the energy cutoff. Although it is straightforward to estimate the number of radial and axial modes required

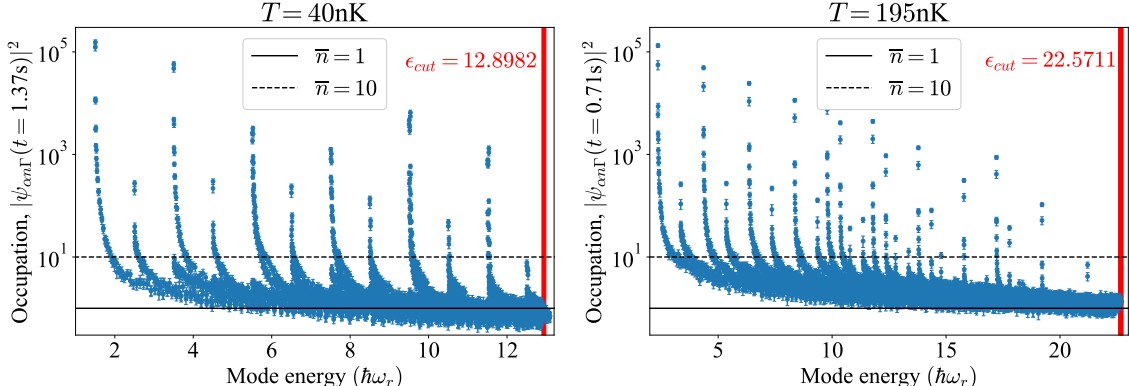

Figure 11: Mean occupation of single-particle modes of the **C** region as a function of mode energy. The occupation is shown at a time part-way through the simulation to ensure some decay events have occurred, for system parameters: (left) $T = 40$nK and $V_b/\mu = 0.78$, and (right) $T = 195$nK and $V_b/\mu = 0.67$. In both cases, the high-energy modes near the cutoff are on the order of $\overline{n} \approx 1$.

to satisfy the condition set by Eq. (23), some care must be taken in choosing the correct number of grid points for the angular grid.

The highest energy angular mode allowed below the energy cutoff can be calculated by assuming all energy is in the angular direction, which gives:

$$|n|_{\max} = \sqrt{\frac{2mr_0^2 \epsilon_{\text{cut}}}{\hbar^2} + \frac{1}{4}}, \tag{29}$$

rounded to the nearest integer. Naïvely, this would suggest that $2|n|_{\max}$ grid points should be used with $n \in [-|n|_{\max}, |n|_{\max}]$. However, the angular component of the single-particle basis are plane waves and are therefore subject to Nyquist aliasing. Formally, aliasing of modes within the **C** region can be avoided by using $n_\theta \geq 4|n|_{\max}$ angular grid points. However, this is typically a large number ($n_\theta \sim 300-400$), which results in restrictive computational requirements for the long-timescale three-dimensional simulations needed for our investigation.

In practice, we choose the grid size such that only a small number of angular modes are subject to aliasing, and only those modes that have a relatively small occupation. We find it is sufficient to use $n_\theta = 150$ in all our simulations. This is confirmed in Fig. 13, which shows that the aliased modes for simulations at $T = 40, 195$nK have an average occupation on the order of 1, and never greater than 10. This is significantly smaller than the occupation of modes of small $n$, which each contain $10^3 - 10^5$ atoms. We have also checked for a full simulation of the experimental procedure at $T = 40$nK that doubling the number of angular modes does not quantitatively change the results of the simulation.

## B.4 Sensitivity of results to barrier width and temperature

We investigate the sensitivity of our results to simulation parameters that are taken directly from the experiment, specifically temperature $T$ and barrier width $w$. As shown in Fig. 14, the decay of the average winding number predicted by simulations does not change significantly as the barrier width $w$ is increased by up to 30%. For much larger values of $w$, the simulations predict that stochastic superflow decay no longer occurs within $t_{\text{hold}} = 1$s. This may be due to the barrier width becoming so large that the dynamics of the superfluid around the barrier maximum are suppressed. However, given that the

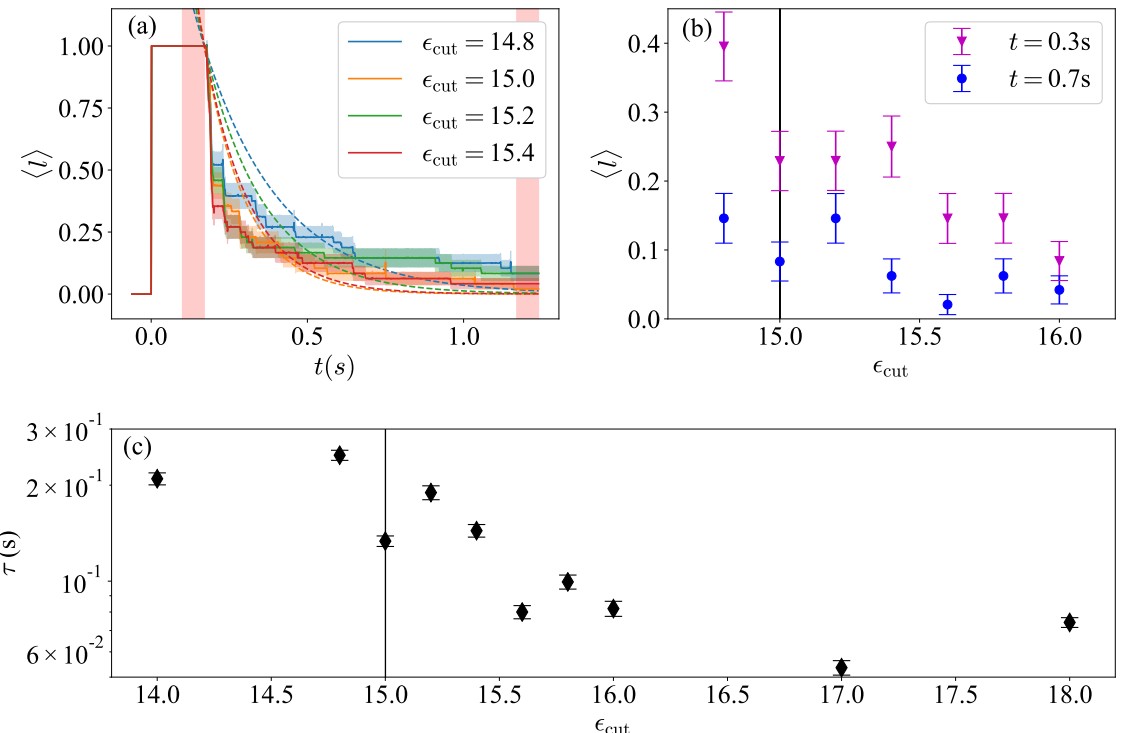

Figure 12: Variation of average winding number and decay timescale as the energy cutoff $\epsilon_{\rm cut}$ is adjusted, for the temperature $T = 85$nK and $V_b/\mu = 0.77$. Here the value of the cutoff $\epsilon_{\rm cut}$ is given in units of $\hbar\omega_r$. (a) The average winding number as a function of time for different values of $\epsilon_{\rm cut}$, where time periods corresponding to the ramping up and down of the barrier are shown in shaded red. Exponential fits to the data are given by the dashed curves. (b) The average winding number at fixed points in time, as a function of energy cutoff. (c) The decay timescale $\tau$, calculated by fitting the trends in (a) to Eq. (15), as a function of $\epsilon_{\rm cut}$. In (b) and (c) the black vertical line gives the precise cutoff value estimated from Eq. (6). The average winding number is obtained by averaging over an ensemble of 48 SPGPE trajectories, with error bars giving a 95% confidence interval (two standard deviations).

width of the atomic density depletion due to the perturbing barrier can be measured with a precision of order $0.1\mu$m, it is unlikely that our estimate of $w$ deviates beyond 10% of its true value.

To investigate the sensitivity of the results on temperature, we consider the experimental data at the hottest temperature, and run simulations across a range of barrier heights at a temperature of $T = 225$nK which is the reported experimental value plus its standard deviation $(T + \sigma_T)$. The comparison of these results to the results for $T = 195$nK is shown in Fig. 15. As one may expect, increasing the temperature reduces the decay timescale at lower barrier heights, improving agreement with the experimental data slightly at those barrier heights. However, for larger barrier heights, we find that increasing the temperature by this amount has very little effect, and thus there is little to no improvement in agreement for the larger barrier heights simulated. Therefore, the precise choice of the simulated temperature within the experimentally quoted confidence interval can be dismissed as a possible origin for the discrepancy observed with the experimental data.

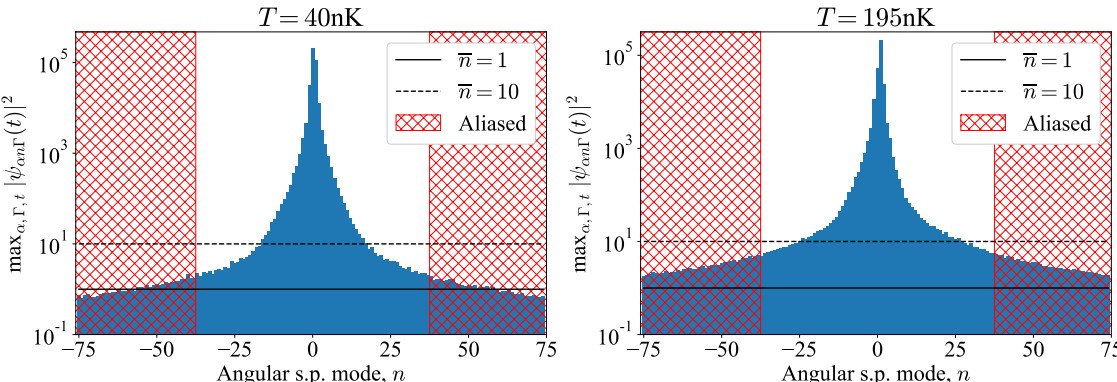

Figure 13: Maximum average occupation of the angular single-particle modes with a given $n$ index, in simulations of the full experimental sequence for temperatures $T = 40\text{nK}$ and $T = 195\text{nK}$, with respective barrier heights $V_b = 0.78\mu$ and $V_b = 0.67\mu$. For a grid with $n_\theta = 150$ points, modes in the red hatched region are formally subject to aliasing. The aliased modes have a maximum occupation on the order of 1, and are never occupied by 10 or more atoms.

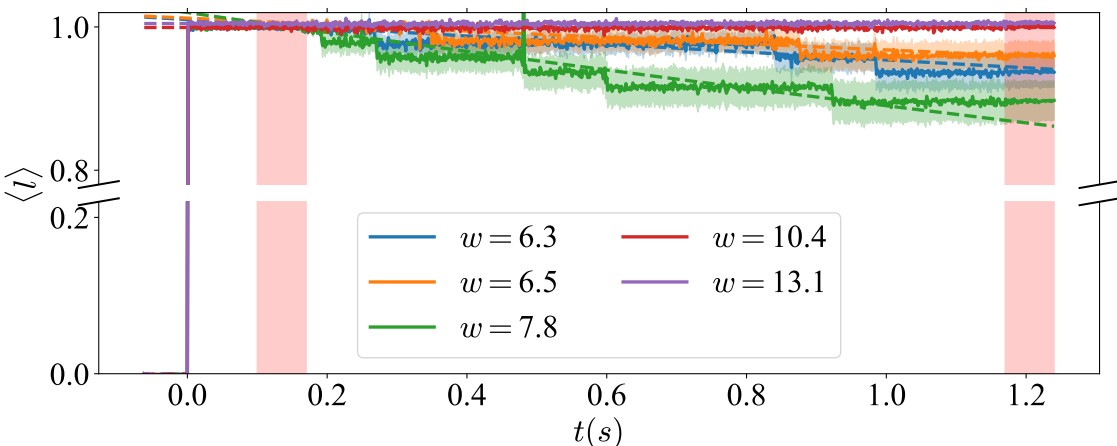

Figure 14: Average winding number as a function of time, for the temperature $T = 195\text{nK}$, and different values of the barrier width $w$, given in microns. The barrier width used for the main results of this paper is $w = 6\mu\text{m}$. Increasing the barrier height slightly (roughly $10 - 20\%$) does not significantly affect the rate of decay, at least not enough to account for the discrepancy with the experiment. More dramatic increases of the barrier width ($\sim 100\%$) result in the suppression of the decay altogether. Shaded regions give a 95% confidence interval for the winding number, which is computed as the ensemble average over 48 trajectories.

## C    Multi-timescale fits of decay

We investigate the effect on our results of fitting the average winding number as a function of time with a multi-parameter exponential model, as opposed to the single timescale model Eq. (15) used in the main body of the work. In Fig. 16 we fit results at each of the four temperatures with models of the form

$$\langle l \rangle = \sum_i c_i e^{-(t - t_2)/\tau_i} \tag{30}$$

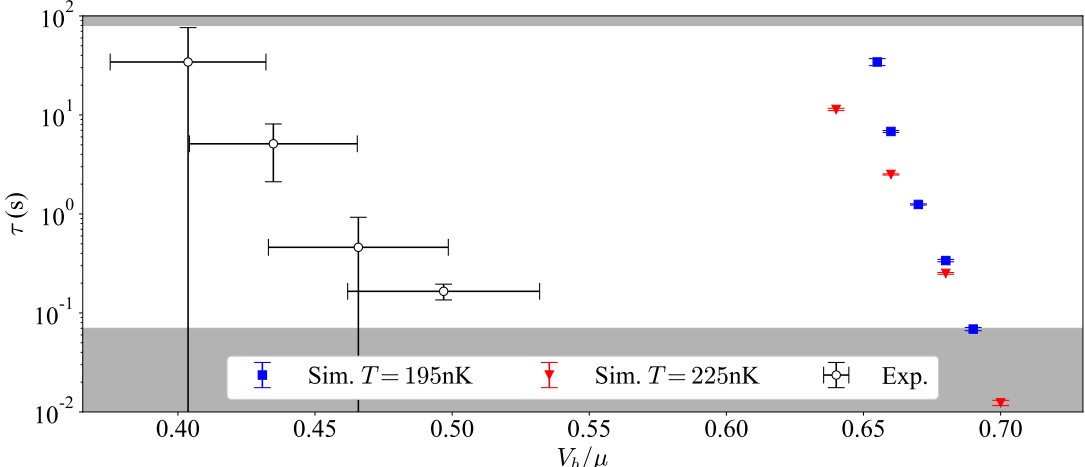

Figure 15: Decay timescale as a function of normalised barrier height, for the experimental results for $T = 195$nK, and simulation results for $T = 195$nK (red diamonds) and $T = 225$nK (blue squares). The temperature $T = 225$nK is the quoted experimental value plus its standard deviation, referring to the upper limit of a 68% confidence interval on the temperature of the experiment. There is a slight quantitative difference between the two simulation trends, with the higher temperature associated with a gentler slope. However, this shift is not sufficient to explain the discrepancy with the experimental data. Error bars shown here represent a 95% confidence interval, with simulation data obtained from an ensemble average over 48 SPGPE trajectories.

where $\{c_i, \tau_i\}$ are free parameters of the fit. In general, a two-timescale fit of the form:

$$\langle l \rangle = c_1 \exp\left(-(t - t_2)/\tau_1\right) + c_2 \exp\left(-(t - t_2)/\tau_2\right) \tag{31}$$

appears to be sufficient to describe the trends for all temperatures. This is with the exception of some $T = 30$nK trends, which clearly have some non-exponential qualities that are not captured by any of these fitting models.

Taking the two parameters $\tau_1, \tau_2$ to represent slow and fast timescales, respectively (we enforce $\tau_1 > \tau_2$), we compare the results of our simulation to the experiment in Fig. 17. This approach does not lead to clear trends, and is challenging to interpret in a meaningful way. In fact, the confidence intervals on many of the two-timescale fitted data points are very large, further demonstrating that a two-timescale fit does not deepen the analysis of the results significantly over the use of a one-timescale fit. Further, there is not a clear physical explanation for why there would be two timescales that would govern the superflow decay mechanism, and thus the addition of additional timescales is entirely *ad hoc*. Finally, the values for the two-timescale fits do not significantly differ from the one-timescale fits, with values spanning the same range of magnitudes as the single-parameter timescales in Fig. 7. Overall, this analysis shows that there is no advantage in using multiple timescale fits over single timescale fits for comparing our simulations to the experiment.

## D  Contribution of energy-damping terms

In our study, we have neglected number-conserving reservoir interaction terms in the SPGPE often referred to as the *scattering* or *energy-damping* terms. These terms are

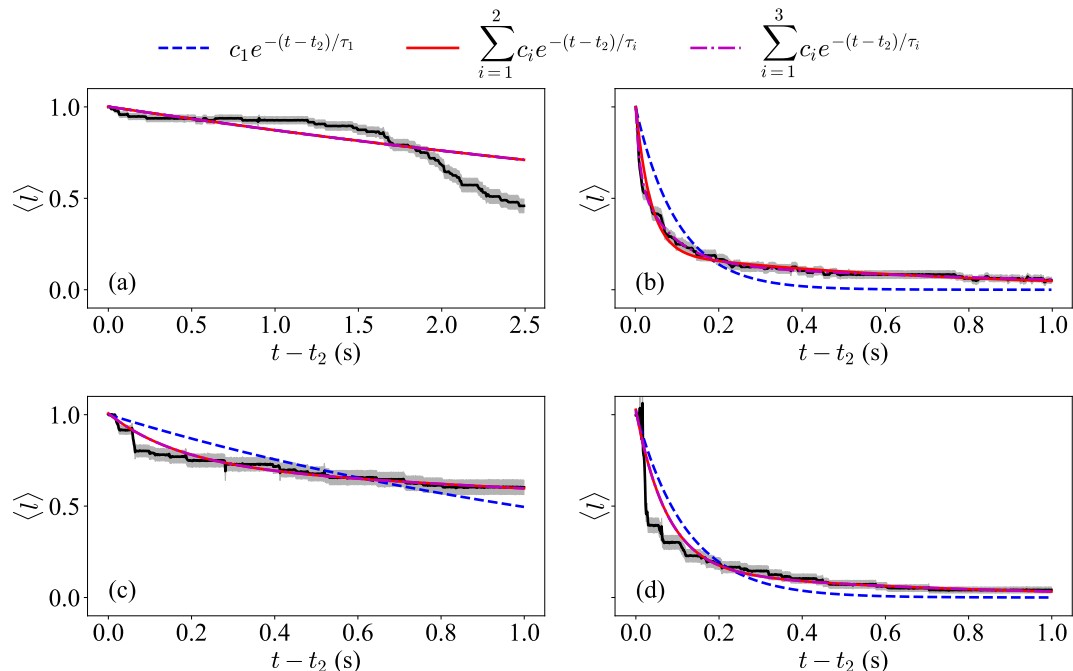

Figure 16: Multi-parameter fits of the average winding number (solid black line) during the period where the barrier is held at its maximum ($t_2$ is the time the barrier first reaches maximum height). Fitting functions are shown at the top of the figure, with fitting parameters $\{c_i, \tau_i\}$. The subfigures refer to four different temperatures and barrier heights: (a) $T = 30$nK, $V_b = 0.7\mu$; (b) $T = 40$nK, $V_b = 0.74\mu$; (c) $T = 85$nK, $V_b = 0.76\mu$; and (d) $T = 195$nK, $V_b = 0.77\mu$. The grey shaded region gives a 95% confidence interval of the average winding number $\langle l \rangle$. In general, the two-timescale fit (red solid line) is sufficient, with the exception of the highly-nonexponential trend in (a).

numerically challenging to implement and are often neglected in SPGPE theory under the justification that they are expected to be dominated by the non-number-conserving $\gamma$ process [41]. Below we quantitatively confirm that their neglect is justified for the simulations performed in this work.

When energy-damping terms are included, the SPGPE is

$$i\hbar d\Psi = \mathcal{P}\big\{(\mathcal{L} - \mu)\Psi dt + i\gamma(\mu - \mathcal{L})\Psi dt + i\hbar d\xi_\gamma(\mathbf{x}, t) \tag{32}$$
$$+ V_\varepsilon(\mathbf{x}, t)\Psi dt - \hbar\Psi dU_\varepsilon(\mathbf{x}, t)\big\}, \tag{33}$$

The energy-damping terms (33) consist of a deterministic evolution term and a noise term, both of which are non-local. The deterministic term describes particle scattering via an effective potential:

$$V_\varepsilon(\mathbf{x}, t) = -\hbar \int d^3\mathbf{y}\,\varepsilon(\mathbf{x} - \mathbf{y})\nabla \cdot \mathbf{j}(\mathbf{y}, t), \tag{34}$$

which is a convolution between the particle current,

$$\mathbf{j}(\mathbf{x}, t) = \frac{i\hbar}{2m}[\Psi\nabla\Psi^* - \Psi^*\nabla\Psi], \tag{35}$$

and the epsilon function

$$\varepsilon(\mathbf{x}) = \frac{\mathcal{M}}{(2\pi)^3} \int d^3\mathbf{k}\,\frac{e^{i\mathbf{k}\cdot\mathbf{x}}}{|\mathbf{k}|}. \tag{36}$$

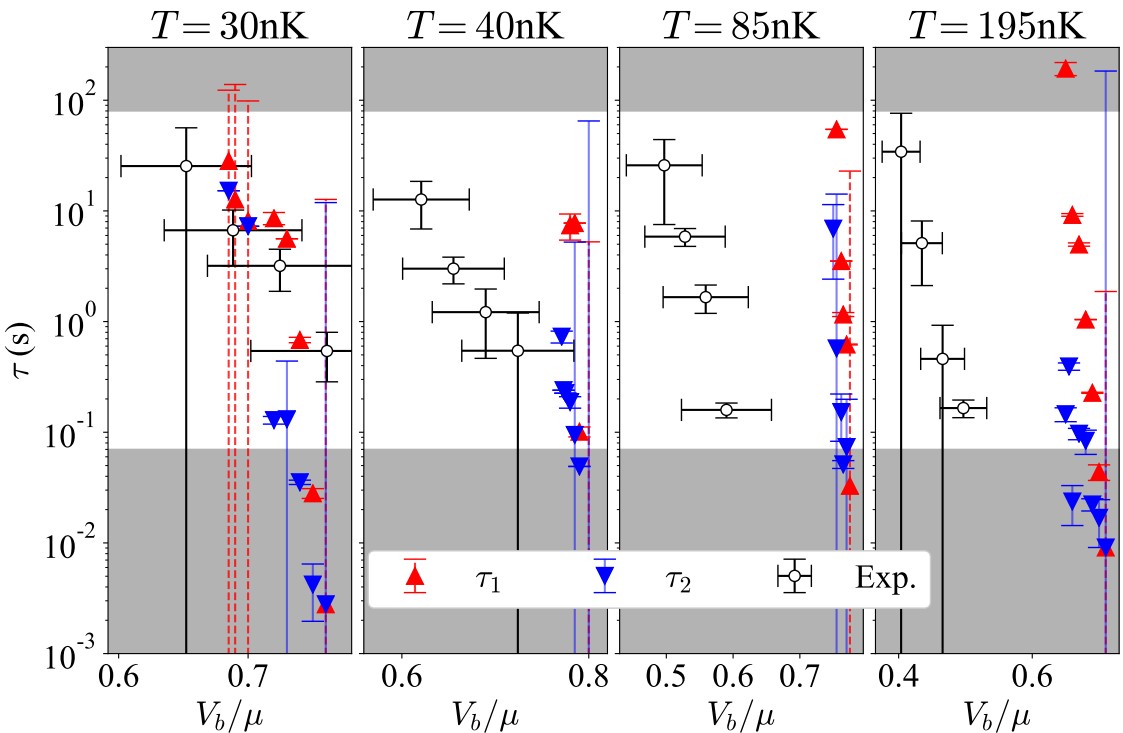

Figure 17: Slow-decay timescale $\tau_1$ (red triangles) and fast-decay timescale $\tau_2$ (blue inverted triangles) as a function of normalised barrier height $\kappa = V_b/\mu$. Error bars give 95% confidence interval. Data points with standard deviation in $\tau$ greater than $10^2$s are excluded from this figure.

The energy-damping noise is real-valued, multiplicative, and non-local in space:

$$\mathbb{E}[dU(\mathbf{x},t)dU(\mathbf{y},t)] = \frac{2k_\mathrm{B}T}{\hbar}\varepsilon(\mathbf{x}-\mathbf{y})dt\,. \tag{37}$$

The strength of the energy-damping terms is captured by the 'energy-damping strength'

$$\mathcal{M} = \frac{16\pi a_s^2}{e^{(\epsilon_\mathrm{cut}-\mu)/k_\mathrm{B}T}-1}\,, \tag{38}$$

which has units of length squared.

To investigate the role of the energy-damping terms, we use an effective two-dimensional form of the SPGPE which assumes that the dynamics in the $z$ dimension are 'frozen'. Explicitly, we assume that the c-field can be factorised as $\Psi(x,y,z,t) = \psi(x,y,t)\phi_0(z)$, where

$$\phi_0(z) = \left(\frac{1}{\pi\sigma_\perp^2}\right)^{1/4}e^{-\frac{z^2}{2\sigma_\perp^2}}\,. \tag{39}$$

By integrating out the $z$ dependence, we arrive at an effective two-dimensional SPGPE that takes the same form as Eq. (32) with the reduced chemical potential, interaction strength, and epsilon functions [83]:

$$\mu_\mathrm{2D} = \mu - \frac{m\omega_z^2\sigma_z^2}{4} - \frac{\hbar^2}{4m\sigma_z^2}\,, \tag{40}$$

$$g_\mathrm{2D} = \frac{g}{\sqrt{2\pi}\sigma_z}\,, \tag{41}$$

$$\varepsilon_\mathrm{2D}(\mathbf{x}) = \frac{\mathcal{M}}{(2\pi)^3}\int d^2\mathbf{k}\, e^{\frac{|\mathbf{k}|^2\sigma^2}{4}}K_0\left(\frac{|\mathbf{k}|^2\sigma^2}{4}\right)e^{i\mathbf{k}\cdot\mathbf{x}}\,. \tag{42}$$

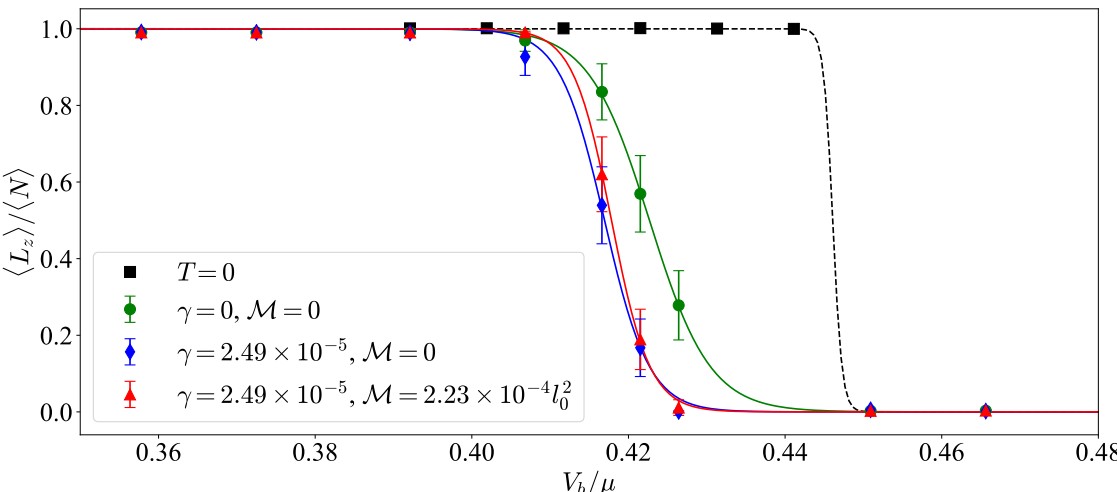

Figure 18: Final value of the average angular momentum per particle (which is equivalent to the final winding number) for the temperature $T = 195$nK and a barrier hold time of $t_{\text{hold}} = 2.5$s. There are several data-sets corresponding to $T = 0$ GPE simulations (black squares), and various subtheories of the SPGPE. Note that $\mathcal{M}$ is given in units of $l_0^2$, where $l_0 = \sqrt{\hbar/m\omega_r} \approx 1.3\mu$m is the the radial lengthscale set by the trap. Each data set is fitted to a sigmoidal function: $\langle L_z \rangle/\langle N \rangle = [\exp((V_b/\mu - \alpha)/\beta) + 1]^{-1}$, where $\alpha, \beta$ are the fitting parameters. Error bars give a 95% confidence interval in the data. Notably, the absence of energy-damping ($\gamma \neq 0, \mathcal{M} = 0$) does not deviate significantly from the full SPGPE result ($\gamma \neq 0, \mathcal{M} \neq 0$).

Here $K_0(\mathbf{x})$ is the zeroth order modified Bessel function. Given the experiment is well within the Thomas-Fermi regime, and $\omega_z$ is not so large as to 'freeze' interactions in the transverse dimension, we use Thomas-Fermi radius to estimate the transverse lengthscale for dimensional reduction $\sigma_\perp = \sqrt{2\mu/m\omega_z^2}$.

An efficient numerical implementation of the energy damping terms for a harmonic trap is detailed in [68]. This algorithm can be adapted for the single-particle basis described in Appendix A. Briefly, the matrix elements for the energy-damping terms are constructed in $k$-space where they are local. For a harmonic trap, the transformation to $k$-space can be achieved using the property that the single-particle modes are eigenfunctions of the Fourier transform. For the toroidal trap, we use a family of Hankel transforms to construct the $k$-space energy-damping matrix elements in cylindrical coordinates $(r, \theta, z) \to (k_r, k_\theta, k_z)$.

The final value of the angular momentum per particle after the simulation sequence for $t_{\text{hold}} = 2.5$s as a function of barrier height is shown in Fig. 18 for $T = 195$nK[6]. To clarify the roles of the various interactions in the SPGPE, we have omitted the inclusion of quantum fluctuations in the initial state, and included a $T = 0$ comparison[7]. We find little quantitative difference when the energy-damping terms are included from the SPGPE as opposed to when they are excluded ($\mathcal{M} = 0$). This justifies our neglect of the contribution of the energy-damping terms in the main results of this work.

---

[6]The parameters for the two-dimensional simulations of the $T = 195$nK experiment are $\mu_{2D} = 6.66\hbar\omega_r$, $\epsilon_{cut} = 17.6\hbar\omega_r$, $\sigma_\perp = 1.23l_0$, $\gamma = 2.49 \times 10^{-5}$, $\mathcal{M} = 2.23 \times 10^{-4}l_0^2$, and a single-particle grid size of $n_r \times n_\theta = 18 \times 200$. Here $l_0 = \sqrt{\hbar/m\omega_r} \approx 1.3\mu$m is the the radial lengthscale set by the trap.

[7]For the $T = 0$ comparison, the initial state of the simulation is the interacting ground state of the GPE, found by evolving the simple-growth SPGPE for 100 trapping periods with no noise term. This is then evolved with the PGPE.

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
