# Peer review of "Superflow decay in a toroidal Bose gas: The effect of quantum and thermal fluctuations"

_SciPost Physics_

## Round 1 · Referee Report · Anonymous (Referee 2) · 2021-6-17

Report

Authors claim that their method is applicable to study phase slip effects over, practically, the whole range of temperatures. I am very skeptical of this claim. Apparently, authors are not familiar with the long list of publications addressing the issue of phase slips in persistent currents, where the dominant mechanism is creation of instanton pairs at low temperatures. Thus, I cannot recommend this paper for publication in its present form. Authors should clearly formulate the domain of validity of their simulations and bring it in line with current understanding of the universal mechanism of phase slips effects. To help along this line, I recommend reading the following papers (and references therein): 1. Kane & Fisher, PRL 68, 1220(1992); 2. Kashurnikov et al, PRB 53, 13091 (1996); 3. Kagan et al, PRA, 61, 045601 (2000); 4. Khlebhikov, PRA 71, 013602 (2005); PRB 77, 014505 (2008); PRB 94, 064517 (2016); PRL 93, 090403(2004).

  • validity: -
  • significance: -
  • originality: -
  • clarity: -
  • formatting: -
  • grammar: -

Author:  Zain Mehdi  on 2021-08-10  [id 1652]

(in reply to Report 2 on 2021-06-17)
Category:
remark
reply to objection

Ultimately, we refute the central claim of the referee that our work is not suitable for publication in SciPost. Our work provides the first beyond-mean-field simulations of the experiment of Kumar et al., and offers a valuable comparison between the experimental results, which are currently lacking theoretical explanation, and a well-established theoretical method for modelling non-equilibrium dynamics of finite-temperature Bose gases. The value of our work can be supported by the comments of Referee 1, who stated “This is a nice work, with results that I believe will be of genuine interest to the ultracold atom community”. Below we address the Referee’s comments in detail.

Our numerical model is based on classical field methodology which, as described in the manuscript, has a domain of validity that is well established by a broad body of theoretical and empirical work in ultracold Bose gases spanning two decades. As outlined in detail in Section 3.1 of our manuscript, classical field methods are the premier technique to use for this analysis, and have been shown to quantitatively describe ultracold Bose gases in both far-from-equilibrium and finite temperature scenarios. The referee is incorrect to suggest they are not valid for the study presented in this work. For example, classical field methods have been used to study phase slips in ultracold Bose gases [e.g. PRA 90 023604 (2014), EPJ D 74 86 (2020)], and have modelled dissipative vortex dynamics in quantitative agreement with experiment [PRA 88 063620 (2013)].

Although we believe phase slips are likely a potentially dominant mechanism for superflow decay, they are only one of several possible mechanisms; other mechanisms include compressible excitations, for example. Crucially, our model is agnostic to the exact source of dissipation. Indeed, the purpose of our study is to model the experiment itself, and compare the experimental results to theoretical predictions within the well-established framework of c-field theory. This is clearly outlined in the introduction, to which we have added additional remarks for further clarity.

The papers that the referee has listed are largely irrelevant to our work - these papers look at a variety of systems (almost all of them one-dimensional, and many of them not cold-atoms based) with models that are highly unsuitable for modelling the non-equilibrium dynamics of a three-dimensional ultracold Bose gas. For the system we consider in our work, one-dimensional models are entirely inappropriate as transverse degrees of freedom are a fundamental source of excitation. The few papers from this list that are relevant to Bose gases use perturbative models in the long-wavelength limit [PRL 93 090403 (2004), PRA 71 013602 (2005), PRA 61 045601] which again do not apply to the highly out-of-equilibrium, long duration 3D dynamics of superflow decay studied in our work.

---

## Round 1 · Referee Report · David Feder (Referee 1) · 2021-6-17

Strengths

(1) This is a well-written, organized, and easy to follow manuscript;
(2) The scientific techniques employed are nover, and the results are interesting and are likely correct.
(3) The right amount of detail is provided.

Weaknesses

There are a few technical points that are rather key to the validity of the work that need to be better justified, as enumerated in the detailed comments below.

Report

This is a nice work, with results that I believe will be of genuine interest to the ultracold atom community. The authors employ a reasonable extension of a powerful but tractable numerical method used to simulate the coupled dynamics of Bose-Einstein condensates and thermal clouds at finite temperatures T < Tc in a fully three-dimensional geometry. But (of course) I have several technical questions and comments. I recommend that it be published in SciPost, once the authors address technical points enumerated below.

Requested changes

(1) In the details of the experiment, the barrier was raised over 70 ms - was this to ensure adiabaticity somehow? If so, adiabatic with respect to what? Does this minimize excitations? What might the numerics say about the important of this timescale?

(2) The experiments lasted for ~7 s. What sets this limit? I assume that it is atom loss, but is this total atom loss from the trap (say from three-body recombination), or heating / evaporation of the BEC into the thermal cloud (say by Oort cloud collisions or other energy / noise source)? Was this a timescale that one can probe with these numerical techniques?

(3) pp. 4-5 it is stated: "The measurement works as follows: if the BEC is in an l = 1 persistent current state when it is released from the trap, then the `hole' in the atomic density at the center of the cloud remains even as the cloud falls and expands. Consequently, the atomic cloud does not significantly overlap with the reference disk of atoms, yielding no measureable interference. In contrast, if the superflow had decayed to the l = 0 state, the hole in the atomic cloud fills during free expansion, resulting in an interaction between the released atoms and the reference disk, and therefore a measureable interference pattern." These statements are problematic, and seem to suggest a conflation of winding number with superfluid circulation and the (quantized) angular momentum. In a BEC with one or a few dark solitons (or similar defects), the phase might wind slowly around the annulus, but jump quickly over short distances in such a way that the total phase might wind 2pi over the full circumference, without needing much superfluid velocity on average (which is the key quantity calculated via Eqs. 9 and 10 in the manuscript). Furthermore, a tiny change in the depth of the defect (say by exciting atoms into the thermal cloud) can lower the phase change across it, and change the circulation from l = 1 to l = 0 without appreciably altering the overall superfluid current. This means that there needn't be any significant angular momentum barrier at the center of the cloud after its release, even for l = 1 states. Instead, the interference between such a state and the reference would show a phase kink but no central hole. Given the way that the circulation is effected (by inserting a moving barrier), I might expect that a dark soliton would naturally form near the barrier region, enforcing the circulation condition without necessarily generating much current. Its decay also wouldn't give rise to much current. All of this to say that I am skeptical of the key observable used in this paper for the decay of superflow: only a small subset of l = 1 states would be expected to behave in the way described.

(4) Above Eq. (6), it is stated "This energy cutoff is typically chosen such that the highest-energy single-particle modes contained within the C region have an occupation of roughly \overline{n}_cut ~ 1-10", but below the equation it is stated that the calculations fix "\overline{n}_cut = 1". This seems like too low a number, given the expectation of higher occupations associated with the BEC. Why do I trust that this is sufficient?

(5) The numerical results presented in Fig. 5 show an extreme sensitivity to barrier height for low temperatures. Is this qualitatively consistent with the experiments? At the same time, my understanding is that the actual quantitative critical barrier heights found numerically aren't close to the experimental values. Do the authors understand what is going on here?

(6) Conversely, the correspondence between numerics and experiment presented in Fig. 7 shows a much closer correspondence between theory and experiment at low temperatures than at high temperatures, where there appears to be a systematic error that increases with temperature. There is extensive discussion following these results that describe attempts to "quantify the disparities" but I didn't see a physical mechanism at play. Generally, points (5) and (6) here would strongly benefit from some analytical work that would at least make an attempt at explaining the most salient features of the numerics.

(7) Section 5.1 describes some limitations of the theoretical model underpinning the numerics for the regime studied, i.e. for temperatures below ~Tc/2. The authors have done a nice job of pointing out the various deficiencies of using the technique in this regime, and it is possible that these limitations account for the discrepancies observed between computation and experiment. But, in the absence of analytical estimates, this is hard to judge. And the calculations don't appear to have been internally benchmarked to know which of these mitigating factors might be the culprit(s). For example, in Sec. 5.1.1, it is stated "it is possible this truncation discards important reservoir interactions...". OK, but is it possible to add (say) one higher-order term and probe the results? The authors state without justification that this would be difficult to implement numerically - why? Likewise for other approximations made. Are any of these approximations controlled enough that one can set reliable convergence criteria? I'll put it another way: what kinds of numerical tests can be performed on related systems whose dynamics are already well-understood, to convey confidence in these results?

  • validity: good
  • significance: high
  • originality: high
  • clarity: top
  • formatting: perfect
  • grammar: perfect

Author:  Zain Mehdi  on 2021-08-10  [id 1651]

(in reply to Report 1 by David Feder on 2021-06-17)
Category:
remark
question
answer to question

We thank the Referee for their detailed report and the opportunity they have provided to improve our work. We are pleased that they recommend publication in SciPost contingent on a satisfactory response to their technical questions, which we address in detail below.

(1) In the details of the experiment, the barrier was raised over 70 ms - was this to ensure adiabaticity somehow? If so, adiabatic with respect to what? Does this minimize excitations? What might the numerics say about the important of this timescale?

(R1) Although the experimental report and its supplemental materials do not justify their precise choice of 70ms barrier raising time, we suspect this may be the upper limit of raising speed achievable while keeping the induced oscillations in angular momentum well below one quanta of angular momentum per particle. Raising the barrier too quickly can excite the superfluid to the next quantised circulation state (l=2), which is certainly unwanted. On the other hand, raising the barrier too slowly could result in undesirable decay events during the raising time. This non-adiabaticity requirement is evidenced by simulations performed in [PRA 90, 023604 (2014)], which show that the raising of the barrier leads to oscillations in the angular momentum that are eventually damped out by the thermal cloud. We observe a similar effect in the simulations we performed in this work. We have added a comment clarifying this requirement in Section 2.

(2) The experiments lasted for ~7 s. What sets this limit? I assume that it is atom loss, but is this total atom loss from the trap (say from three-body recombination), or heating / evaporation of the BEC into the thermal cloud (say by Oort cloud collisions or other energy / noise source)? Was this a timescale that one can probe with these numerical techniques?

(R2) The experimental report quotes a ~27s lifetime of the experiment due to atom loss, but does not elaborate on the dominant mechanism for this loss. Typically, it is difficult to determine the mechanism of atom loss empirically. However, the experiment is based on the apparatus first described by Ramanathan et al. [PRL 106 130401 (2011)] who state the ~30s 1/e lifetime of the condensate is vacuum limited. This would suggest one-body loss was the dominant limitation for the condensate lifetime in that apparatus, and it is reasonable to assume this is the same limitation for this experiment. Furthermore, we estimate that three-body loss is vanishingly small for the experimental parameters, so three-body loss can be ruled out as the dominant mechanism.

Modelling the full 7-second sequence is likely beyond the capacity of our numerical methods - both in terms of computational expense (as many trajectories are also required), as well as in minimising the numerical error that accumulates with simulation time. The timescales probed in our simulations (~1s-2.5s depending on the speed of the decay) were chosen to be sufficiently long such that the longest decay timescales observed are as large as the longest timescales measured in the experiment. There would be little point in simulating beyond this timescale for the purposes of comparing to the experimental data. We have modified our discussion in Section 4.2 to include this point of clarification.

(3) pp. 4-5 it is stated: "The measurement works as follows: if the BEC is in an l = 1 persistent current state when it is released from the trap, then the `hole' in the atomic density at the center of the cloud remains even as the cloud falls and expands. Consequently, the atomic cloud does not significantly overlap with the reference disk of atoms, yielding no measureable interference. In contrast, if the superflow had decayed to the l = 0 state, the hole in the atomic cloud fills during free expansion, resulting in an interaction between the released atoms and the reference disk, and therefore a measureable interference pattern." These statements are problematic, and seem to suggest a conflation of winding number with superfluid circulation and the (quantized) angular momentum. In a BEC with one or a few dark solitons (or similar defects), the phase might wind slowly around the annulus, but jump quickly over short distances in such a way that the total phase might wind 2pi over the full circumference, without needing much superfluid velocity on average (which is the key quantity calculated via Eqs. 9 and 10 in the manuscript). Furthermore, a tiny change in the depth of the defect (say by exciting atoms into the thermal cloud) can lower the phase change across it, and change the circulation from l = 1 to l = 0 without appreciably altering the overall superfluid current. This means that there needn't be any significant angular momentum barrier at the center of the cloud after its release, even for l = 1 states. Instead, the interference between such a state and the reference would show a phase kink but no central hole. Given the way that the circulation is affected (by inserting a moving barrier), I might expect that a dark soliton would naturally form near the barrier region, enforcing the circulation condition without necessarily generating much current. Its decay also wouldn't give rise to much current. All of this to say that I am skeptical of the key observable used in this paper for the decay of superflow: only a small subset of l = 1 states would be expected to behave in the way described.

(R3) We agree our explanation of the experimental measurement was problematic, since we erroneously stated that the measurement relies on whether or not there is a persistent `hole’ in the dropped atomic cloud in its time of flight (TOF). We have corrected our explanation in the main text, which we summarise below.

In the experiment of Kumar et al., there is always interference between the atomic cloud and the reference disk during TOF expansion, yielding an interference pattern. This interference pattern is used to reconstruct the phase profile around the annulus [PRL 113 135302 (2014)], and can be used to explicitly determine the winding number - the precise details of this method is a subject of an earlier work of the experimental group [PRX 4 031052 (2014)]. Therefore, the winding number we compute is precisely the same quantity measured in the experiment, justifying its use for the comparison in this work.

As a separate note, we agree that a defect such as a vortex crossing the loop along which the winding number is calculated could lead to a discontinuous phase change and reduce the overall circulation. In fact, in our calculation of winding number as a function of time, we find that it is not uncommon for the winding number to drop to zero for a very brief period of time, which we can attribute to a discontinuity in the phase in the path of the loop - this is due to the density going to zero at some point in the loop, and the velocity field thus becoming undefined. However, we do not observe such effects lasting for any significant duration on the timescales of both the experiment and the presented simulations. This is likely because vortices have a very short lifetime (relative to seconds) due to dissipation from, for example, the thermal cloud.

As a final comment, we have found that the average winding number (computed as explained in our manuscript) empirically agrees with the angular momentum per particle after the barrier is lowered. As the experimental measurement of winding number was performed in TOF after the barrier is lowered, it can be safely interpreted as a measurement of quantised angular momentum. We believe the reason for this is that the quantised circulation states are the only excitations that are persistent on the long timescales of the experiment.

(4) Above Eq. (6), it is stated "This energy cutoff is typically chosen such that the highest-energy single-particle modes contained within the C region have an occupation of roughly \overline{n}_cut ~ 1-10", but below the equation it is stated that the calculations fix "\overline{n}_cut = 1". This seems like too low a number, given the expectation of higher occupations associated with the BEC. Why do I trust that this is sufficient?

(R4) The C region contains all modes that have occupation of 1 and greater. Therefore, all the highly occupied modes associated with a BEC are well represented in this region. Furthermore, the sparsely occupied modes outside of the C region are also included through the SPGPE damping terms associated with the thermal cloud. We have made a correction in the manuscript around Eq.(6) to clarify this.

This choice of cutoff is standard within classical field studies, and is often not rigorously justified. In our work we explicitly check the validity of the cutoff in Appendix C by demonstrating that the decay timescale (the key computed quantity) does not vary significantly as the cutoff is varied around the chosen value.

(5) The numerical results presented in Fig. 5 show an extreme sensitivity to barrier height for low temperatures. Is this qualitatively consistent with the experiments? At the same time, my understanding is that the actual quantitative critical barrier heights found numerically aren't close to the experimental values. Do the authors understand what is going on here?

Qualitatively, the experiment also shows extreme sensitivity to barrier height at low temperatures. This is most clearly shown in Figure 7, where we see some level of agreement between the experimental and theoretically estimated timescales for the lowest temperature. In fact, the experimental results show an extreme sensitivity to barrier height at all temperatures, since tuning the barrier height over only ~10% of the chemical potential results in order of magnitude changes in the decay timescale.

(R5) Quantitatively, however, we find that for the two intermediate temperatures the theoretical model predicts a much higher sensitivity to small variations in the barrier height than the experiment (quantified in Figure 8). Although we are uncertain of the origin of the discrepancy, we can rule out a number of sources, and have documented our reasoning for doing so in the manuscript. Effects we rule out are: shot-to-shot atom number fluctuations (Sec. 5.2.1), mismatch between simulation and experimental parameters (Appendix B), choice of fitting function (Appendix C), and number-conserving reservoir interactions (Appendix C). As we speculate in Sec 5.1.1, this discrepancy may be partially due to the neglect of terms in our model that become non-negligible at the intermediate temperatures. However, the discrepancy may also be due to additional physical processes in the experiment not included in our simulations (such as temporal variation in the induced potentials or stray fields), which might increase the rate of superflow decay.

We have clarified the discussion of Figure 5 to note the qualitative and quantitative comparison with the experiment, and flag further analysis in the following section

(6) Conversely, the correspondence between numerics and experiment presented in Fig. 7 shows a much closer correspondence between theory and experiment at low temperatures than at high temperatures, where there appears to be a systematic error that increases with temperature. There is extensive discussion following these results that describe attempts to "quantify the disparities" but I didn't see a physical mechanism at play. Generally, points (5) and (6) here would strongly benefit from some analytical work that would at least make an attempt at explaining the most salient features of the numerics.

(R6) There have been several works that have attempted to use analytic models to understand persistent current formation and decay for the considered experiment and related systems, which we summarise below.

Analytic work related to the considered experiment In the experimental report of Kumar et al [PRA 95 021602 (2017)], the authors consider an analytic model of a solitonic vortex to estimate the energy barrier that couples the circulating and non-circulating states. Within this model, they compute a quantum tunnelling rate and a thermal activation rate using an Arrhenius law. Neither model accounts for the experimentally observed decay; the former is negligible for the experimental parameters, and the latter demonstrates different qualitative scaling with temperature. Kunimi et al. [PRA 99 043613 (2019)] perform a more detailed semi-analytic calculation within mean-field theory, where they use numerical diagonalisation to determine the energy barrier and an Arrhenius law to estimate the decay rate due to thermal activation. They found that the estimated decay rates are negligible compared to experimental data (with discrepancies ranging between 50-200 orders of magnitude). Other related analytic work Snizhko et al [PRA 94 063642 (2016)] derive a finite-temperature Arrhenius-like law for thermally activated phase slips due to a rotating barrier in a toroidal BEC. Although this model is non-perturbative, calculations are one-dimensional and significantly overestimate the critical rotation frequency observed in experiment [PRL 110 025302 (2013)]. Cominotti et al [PRL 113 025301 (2014)] consider a 1D Luttinger liquid model to probe different regimes of interaction strengths and barrier heights for persistent currents. This model is beyond mean-field and useful for characterising a wide range of regimes, but is fundamentally perturbative due to the low-energy formulation within a Luttinger liquid framework. Perez-Obiol et al [PRA 102 063302 (2020)] analytically estimate the critical velocities and energy spectra of a stirred toroidal BEC. This model is useful for understanding persistent current production for a range of interaction strengths, but only qualitatively as it is formulated within 1D mean-field theory.

In essence, none of the analytic approaches above are suitable to support a quantitative analysis of the considered experiment. In general, analytical approaches are either perturbative, restricted to one-dimension and/or restricted to mean-field theory. It is in some ways unsurprising that these approaches are unsuitable for the study of the considered experiment, which is a three-dimensional, highly non-equilibrium system where beyond-mean field effects are important. This is precisely the regime where c-field methods are known to work best (see 3.1 of the main text).

Although analytical methods can determine the scaling of the underlying mechanisms, the aim of this work is to perform detailed, first-principles numerical modelling of the experiment. We have clarified this point in our introduction, and expanded our literature review to include previous attempts at modelling the experiment that are described above.

(7) Section 5.1 describes some limitations of the theoretical model underpinning the numerics for the regime studied, i.e. for temperatures below ~Tc/2. The authors have done a nice job of pointing out the various deficiencies of using the technique in this regime, and it is possible that these limitations account for the discrepancies observed between computation and experiment. But, in the absence of analytical estimates, this is hard to judge. And the calculations don't appear to have been internally benchmarked to know which of these mitigating factors might be the culprit(s). For example, in Sec. 5.1.1, it is stated "it is possible this truncation discards important reservoir interactions...". OK, but is it possible to add (say) one higher-order term and probe the results? The authors state without justification that this would be difficult to implement numerically - why? Likewise for other approximations made. Are any of these approximations controlled enough that one can set reliable convergence criteria? I'll put it another way: what kinds of numerical tests can be performed on related systems whose dynamics are already well-understood, to convey confidence in these results?

(R7) We are pleased that the referee is satisfied with our discussion of the limitations of our numerical approach, and we agree it is difficult to judge whether they account for the discrepancies between our numerical modelling and the experiment. We have amended section 5.1.2 of the manuscript to clarify aspects of our discussions of the limitations of our model.

With regards to the referee’s comment on internally benchmarking the numerical approach, the classical field framework used in our work has been extensively benchmarked in equilibrium and non-equilibrium situations across different geometries, atomic species, and temperature ranges. This is both with other numerical methods, and experimental data - see Refs. 49-82 in the revised manuscript. In this sense the validity of our numerical method is well-established by almost two decades of work in the broader literature. This is to say that we would be extremely surprised if the breakdown of the truncated Wigner approximation or the inclusion of additional reservoir interactions could substantially resolve the quantitative disagreement between theory and experiment. Nevertheless, we cannot rigorously rule this out, and therefore speculate that it could potentially contribute to the observed disagreement.

With regards to the referee’s suggestion about adding the next order reservoir interaction, this is potentially an entire research programme on its own, with several reasons why it would be impractical to numerically implement for this work. For one, the numerical method we use to make these large multi-mode stochastic 3D simulations tractable is tailored very specifically for the equations we use. The addition of the next order of the interaction will result in additional nonlinear and stochastic terms, of forms that are not obviously efficient to calculate using our numerical method, at least not without a great deal of work. Although feasible in small one dimensional systems, including higher-order terms in either the reservoir interaction or the Hamiltonian nonlinearity is very challenging numerically for simulations of this scale, and well beyond the scope of the present work.

There is a more fundamental reason why this is not simple to do. If we add the next order reservoir interaction into the master equation, and map the resulting master equation into a PDE for the multi-mode Wigner function, we find that it leads to third (and higher) order derivative terms. That means the PDE cannot be written as a Fokker-Planck equation, and thus cannot be mapped to a stochastic differential equation that can be efficiently simulated. Beyond FPE terms have been treated using: i) Stochastic difference equations that are highly unstable numerically [EPL 56 372 (2001)], and ii) Variational techniques that are only well-tested and practical for quantum systems containing a very small number of modes [PRA 68 053604 (2003)]. These approaches are not practical for modelling the experiment, nor to our knowledge, have ever been applied to modelling large-scale, multimode, finite-temperature Bose-Einstein condensates.

---

## Editorial Decision

resubmitted